# Heat flows enrich prebiotic building blocks and enhance their reactivity

Thomas Matreux[1,3], Paula Aikkila[1,3], Bettina Scheu[2], Dieter Braun[1] & Christof B. Mast[1✉]

The emergence of biopolymer building blocks is a crucial step during the origins of life[1–6]. However, all known formation pathways rely on rare pure feedstocks and demand successive purification and mixing steps to suppress unwanted side reactions and enable high product yields. Here we show that heat flows through thin, crack-like geo-compartments could have provided a widely available yet selective mechanism that separates more than 50 prebiotically relevant building blocks from complex mixtures of amino acids, nucleobases, nucleotides, polyphosphates and 2-aminoazoles. Using measured thermophoretic properties[7,8], we numerically model and experimentally prove the advantageous effect of geological networks of interconnected cracks[9,10] that purify the previously mixed compounds, boosting their concentration ratios by up to three orders of magnitude. The importance for prebiotic chemistry is shown by the dimerization of glycine[11,12], in which the selective purification of trimetaphosphate (TMP)[13,14] increased reaction yields by five orders of magnitude. The observed effect is robust under various crack sizes, pH values, solvents and temperatures. Our results demonstrate how geologically driven non-equilibria could have explored highly parallelized reaction conditions to foster prebiotic chemistry.

The formation of the first biopolymers and their building blocks on the early Earth was a key moment during the origins of life. Possible reaction pathways for the formation of nucleotides, amino acids and lipids have been studied with great success[1–6]. To uncover the details of such pathways and to map them reproducibly, laboratory experiments usually start with well-defined concentrations of previously purified reactants[15–18] (Fig. 1a). Often, a well-defined sequence of manual steps is required, such as the addition of further reactants or the selective purification of intermediates to increase the yield of the final products. Although high starting concentrations can be helpful, the number of side products substantially increases for complex reaction pathways. Therefore, without some form of intermediary purification[19–21], the reaction will result in vanishingly small concentrations of the desired product or even be completely inhibited.

Although such further steps seem artificial and challenging to perform in a prebiotic context, some plausible, substance-specific purification processes have been found, including the crystallization of nucleotide precursors[22,23] or the precipitation of aminonitriles[24]. Other selective approaches use reaction-specific interconversions[25], ultraviolet (UV) light[26], sequestration[27] or adsorption and enrichment of RNA[28] or RNA building blocks[29] on surfaces. Coacervates locally optimize conditions for prebiotic chemistry through phase separation[30,31]. However, these mechanisms only work for specific prebiotic compounds and may require mutually exclusive environmental conditions. A natural mechanism that spatially separates and simultaneously purifies a wide range of prebiotic compounds and implements numerous connected reaction environments remains elusive.

In this work, we show that heat flows through thin rock cracks provide an answer to this problem. These cracks, for example, generated by thermal stress, form large networks and were presumably ubiquitous on the early Earth[9,10]. Specifically, water-filled fractures and thin, connected pathways are found in a variety of geological settings, from mafic to ultramafic rocks in volcanic complexes to geothermal or hydrothermal systems and sedimentary layers, for example, in shallow submarine or lacustrine environments[10,32,33]. Our results show the simultaneous but spatially separated, heat-flux-driven purification of more than 50 prebiotically relevant organic compounds. Although the overall system is fed by slow geothermal fluxes, each rock fracture hosts local solvent convection (Fig. 1b, black arrows) and thermophoretic drift of solutes along the temperature gradient (Fig. 1b, white arrows). The interplay of both effects increases the concentrations differently for the various solutes, shifting their concentration ratios by several orders of magnitude. This specificity results from the sensitive dependence of thermophoresis on charge, size and solvent interaction[8,34–37]. Heat flows are readily available in the early Earth lithosphere, cooling down from accretion and being fed by radioactive decay. Such large-scale thermal gradients are superimposed by local thermal gradients, for example, in volcanically active or geothermal environments, and as a thermo-aureole linked to magmatic intrusions[32]. Also, heat is a waste product of various chemical and petrological processes. This mechanism thus offers a wide range of ubiquitously available reaction conditions for prebiotic chemistry within geological compartments.

Heat fluxes were shown to locally enrich nucleotides[7,38], help copy oligomers in a length-dependent manner[39], generate local pH differences[40,41] and optimize salt conditions for ribozymes[42]. However, it is

[1]Systems Biophysics, Ludwig-Maximilians-Universität München, Munich, Germany. [2]Department of Earth and Environmental Sciences, Ludwig-Maximilians-Universität München, Munich, Germany. [3]These authors contributed equally: Thomas Matreux, Paula Aikkila. ✉e-mail: christof.mast@physik.uni-muenchen.de

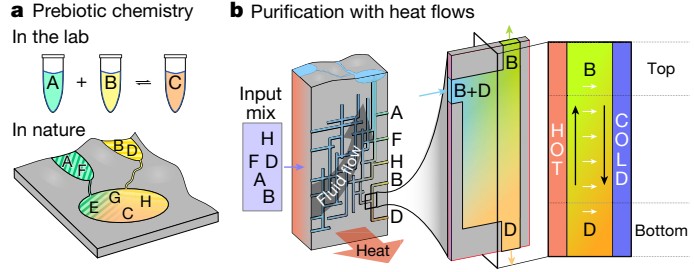

**Fig. 1 | Purifying the prebiotic clutter. a**, Prebiotic chemistry reactions often require precisely timed mixing of well-defined starting materials with intermediate purification steps for high product yields. In nature, starting solutions are complex mixtures that react to produce many undesirable side products. **b**, Ubiquitous heat flows through thin rock fractures, fed by geothermal fluid flow (grey arrow), form a geo-microfluidic system that separates even highly similar prebiotic chemicals from each other through substance-sensing thermophoresis (white arrows) and fluid convection (black arrows). Owing to the geological scale, many different solution compositions are reached simultaneously.

unclear how heat fluxes affect complex mixtures of small, prebiotically relevant organics and whether these can be selectively enriched and separated spatially in geologically plausible systems.

To answer this question, we used geologically inspired microfluidic heat flow chambers as a first step to experimentally study the thermophoretic accumulation of complex mixtures in a single rock fracture. In the second step, we numerically investigated the effects on large fracture networks and experimentally verified the results in proof-of-principle minimal network systems.

Numerical modelling of such networks requires the mostly unknown thermophoretic properties of the molecules involved, which are not accessible at low concentrations using available methods. For instance, high-sensitivity methods rely on fluorescent labels, which substantially alter the diffusive and thermophoretic properties of the target molecule[43]. Holographic techniques require high concentrations that lead to pH shifts and only allow the measurement of one component at a time[44]. We have, therefore, developed a method to simultaneously measure the thermophoretic properties of up to 20 compounds from their mixtures, limited only by high-performance liquid chromatography (HPLC) sensitivity.

Our experimental and numerical results show that even weak heat fluxes separate and locally enrich 2-aminoazoles, amino acids, nucleobases and nucleotides in their various phosphorylation states. The effect distinguishes substances of equal mass and works in a wide pH range and for different solvents. We demonstrate the benefit for prebiotic chemistry with the TMP-driven dimerization of glycine, which is enhanced by selective purification of the reactants in a single heat flow chamber and numerically model the reaction in networks of connected cracks to explain the large-scale effects in natural environments.

We mimicked a single thin rock fracture with a heat flow chamber, defined by a 170-μm-thin fluorinated ethylene propylene (FEP)-defined microfluidic structure between a heated (40 °C) and a cooled sapphire (25 °C) (Fig. 2a and Extended Data Fig. 1; Methods) and filled it with a mixture of prebiotic compounds $i$ (initial concentration $c_0$ = 20–50 μM each). By the interplay of fluid convection and solute thermophoretic drift $v_{T,i} = -\nabla T \times S_{T,i} \times D_i$, an exponential concentration profile built up within the heat flow chamber[45,46], with $S_{T,i}$ denoting the Soret coefficient as a measure of thermophoretic strength and $D_i$ the diffusive mobility of the solute. After 18 h, we stopped the experiment, froze the chamber and divided its contents into four parts of equal volume. The concentration ratio of the respective substances in each section relative to the chamber-averaged concentration $c_0$ of this substance was then

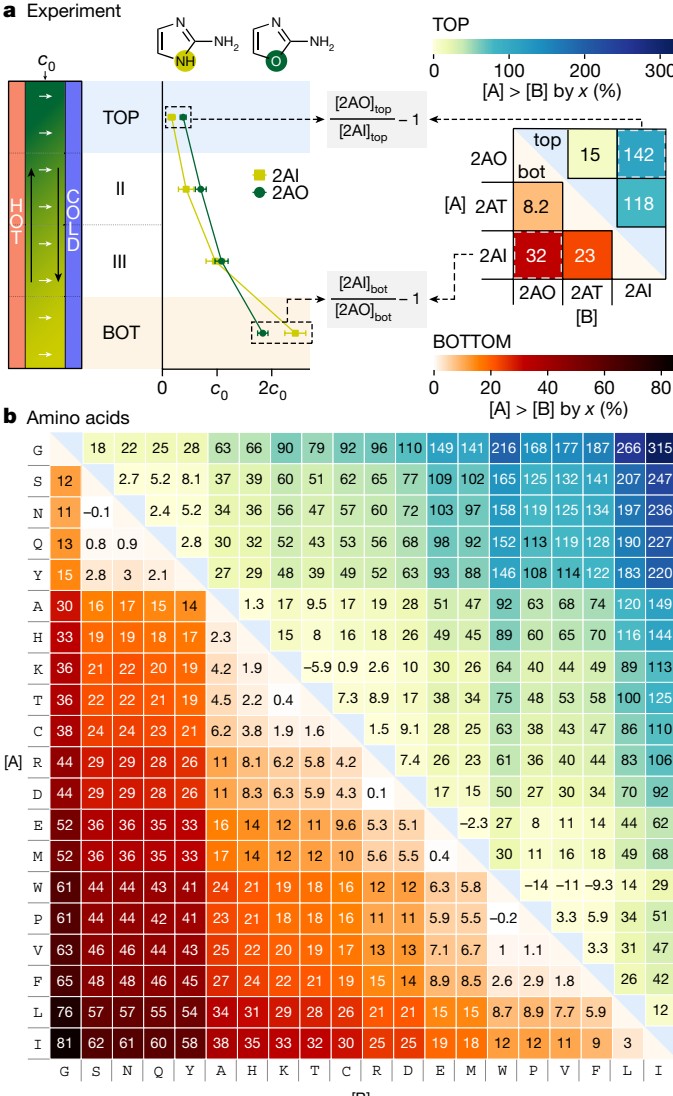

**Fig. 2 | Thermophoretic enrichment of prebiotic organics in a single heat flow chamber. a**, Illustration of the selective enrichment of prebiotic components for a mixture of 2AI, 2AT and 2AO in a thermal gradient (25–40 °C). Heat maps show the concentration ratios of all possible substance pairs in the bottom (orange shaded) and top (blue shaded) sections of the chamber. For example (dashed boxes), in the top section, 2AO is (142 ± 52)% more concentrated than 2AI, whereas in the bottom section, 2AI is (32 ± 3.6)% enriched over 2AO (errors = s.d., three repeats). **b**, Enrichment in a mixture of all proteogenic amino acids (30 μM each) reveals a strong enrichment of aliphatic amino acids isoleucine (I), valine (V) and leucine (L) in the bottom section (orange shade) against glycine (G) (up to (81 ± 25)%) and serine (S), asparagine (N) and glutamine (Q) (up to (62 ± 17)%). Consistently, the aliphatic amino acids are strongly depleted in the top section (blue shade), resulting in up to (315 ± 138)% higher local glycine concentration. See Extended Data Figs. 2–4 for measurements at other initial pH values, temperature gradients, salt concentrations and error maps.

determined by HPLC. In this way, measurements of different mixtures of nucleobases or amino acids could be compared despite different initial concentrations of the individual components. The mean enrichment $[A]_j/[B]_j - 1$ between compounds A and B in the top ($j$ = top) and bottom ($j$ = bot) chamber parts was determined from triplicate experiments (Figs. 2–3, Extended Data Figs. 2–5 and raw data in Supplementary Tables 5–65; Methods).

The degree of enrichment and spatial separation of different substances in a single heat flow chamber is visualized by comparing their

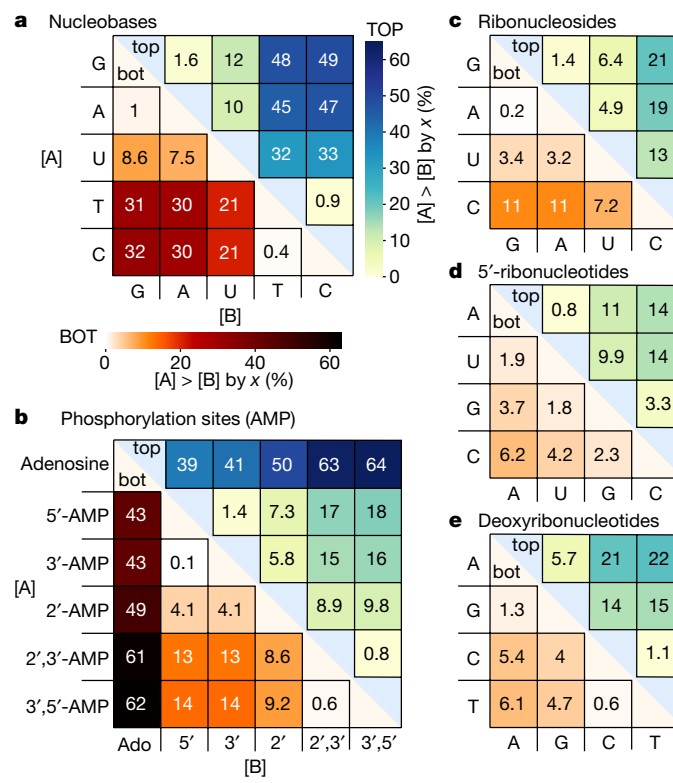

**Fig. 3 | Enrichment of nucleobases, nucleosides and nucleotides. a**, In mixtures of nucleobases A, U, T, C and G with an initial concentration of 30 μM, C and T are enriched up to $(32 \pm 13)\%$ (error = s.d., three repeats) against G, A and U in the bottom section. Enrichments in the top section are inverted but at lower absolute concentrations because of thermophoretic depletion (11.5 to 31 μM, corresponding to 0.5–0.9-fold $c_0$). **b**, Enrichment strongly depends on the phosphorylation state. In the bottom section, cyclic 2′,3′-AMP and 3′,5′-AMP are enriched up to $(62 \pm 0.3)\%$ relative to adenosine and $(14 \pm 0.6)\%$ relative to the linearly phosphorylated 5′-AMP, 3′-AMP or 2′-AMP. The enrichment is inverted in the top section. **c–e**, Enrichment patterns similar to those of nucleobases, but with reduced strength, are also found for nucleosides (**c**) and nucleotides (**d,e**). Extended Data Fig. 5 and Supplementary Figs. 1–3 show more conditions and error tables.

concentrations in the uppermost (blue shaded) and lowermost (orange shaded) sections in Figs. 2 and 3. In a mixture of 2-aminoazoles, simple RNA precursors and catalysts, we found that, in the bottom section, 2-aminoimidazole (2AI) is enriched over 2-aminooxazole (2AO) by $\overline{[2AI]_{bot}}/\overline{[2AO]_{bot}} - 1 = (32 \pm 3.6)\%$. In the diluted top section, 2AO is enriched by $\overline{[2AO]_{top}}/\overline{[2AI]_{top}} - 1 = (142 \pm 52)\%$ over 2AI (Fig. 2a and Extended Data Fig. 2 for errors). The absolute concentrations of species in the bottom section range from 64 to 173 μM, whereas we measured between 6 and 20 μM in the top section. Owing to the almost complete depletion (about 0.17-fold $c_0$) of the thermophoretically stronger species, the remaining, weaker accumulating species are highly enriched here.

Amino acids, the building blocks of peptides and proteins, are a critical class of prebiotically plausible and relevant substances[6,47]. To determine their thermophoretic separation, we analogously investigated a mixture of all 20 proteinogenic amino acids at physiological pH (Fig. 2b). Although thermophoretically similar amino acids such as aspartic acid (D) and arginine (R) are not strongly separated from each other ($\overline{[D]_{bot}}/\overline{[R]_{bot}} - 1 = (0.1 \pm 2.8)\%$), we found a massive separation of isoleucine (I) against glycine (G) by $\overline{[I]_{bot}}/\overline{[G]_{bot}} - 1 = (81 \pm 25)\%$ in the bottom fraction. Sorting the amino acids according to their mutual enrichment, we found that glycine, as the smallest amino acid, concentrates most weakly in the bottom fraction, whereas the aliphatic

amino acids leucine (L), valine (V) and I accumulate most strongly. The situation is inverted in the top fraction, as the amino acids with the weakest thermophoretic strength have the highest concentration here ($\overline{[G]_{top}}/\overline{[I]_{top}} - 1 = (315 \pm 138)\%$). Even mass-identical amino acids I and L are separated up to $\overline{[L]_{top}}/\overline{[I]_{top}} - 1 = (12 \pm 0.3)\%$. The enrichment values and their errors for amino acids at other pH values and different ionic strengths show similar separation patterns (Extended Data Figs. 3 and 4). In a mixture of 17 proteogenic and nine non-proteogenic amino acids, we found no clear bias towards either (Extended Data Fig. 2). We also found no thermophoretic separation between amino acids[48] and nucleosides of different chirality (Extended Data Fig. 5s−u).

In a mixture of all canonical nucleobases in water, thymine (T) and cytosine (C) are enriched by up to $(32 \pm 13)\%$ over adenine (A) and guanine (G) in the bottom section of the chamber (Fig. 3a and Extended Data Fig. 5a for errors). The observed pattern is mainly identical in 10% formamide[16] solution, in 100 mM phosphate buffer[15,17,18,22] and for various crack diameters and pH values (Extended Data Fig. 5b,c,e−m). In 10% methanol[21,24,29] solution, accumulation and enrichment are almost entirely suppressed (Extended Data Fig. 5d).

Adenosine nucleotides (AMPs) are concentrated between $(43 \pm 1)\%$ and $(62 \pm 0.3)\%$ more than the adenosine nucleoside (Ado) in the bottom section, owing to the extra phosphate group that increases the charge and, thus, the thermophoretic strength[35] (Fig. 3b). Cyclic AMPs are accumulated up to $(14 \pm 0.6)\%$ more than linear AMPs. Despite identical mass, 2′-AMP is enriched by $(4.1 \pm 0.3)\%$ over 3′-AMP and 5′-AMP. Similar results are found for cytidine and its nucleotides (Extended Data Fig. 5q).

Mixtures of RNA or DNA nucleosides and nucleotides in all phosphorylation states show a similar enrichment pattern as for bases, albeit with lower magnitude (Fig. 3c−e and Extended Data Fig. 5n−p). The reduced thermophoretic separation is reasonable because the ribose and phosphate groups added to the different bases are identical for all nucleosides and nucleotides, thus decreasing the relative structural difference between species.

Because thermogravitational accumulation approaches exponential concentration profiles in the steady state[45,46], the volume-averaged enrichments measured are lower than the actual values present at the bottom. For instance, separation in the lowest one-twelfth is substantially higher compared with the lowest quarter, at the cost of an increased error (Extended Data Fig. 2b). At the lowest end of the exponential profile, 2AI is enriched up to 406% over 2AO (Extended Data Fig. 2c, $S_{T,i}$ from Extended Data Table 1), in contrast to $(32 \pm 3.6)\%$ in the lowest quarter (Fig. 2a). See Extended Data Fig. 2e for how this translates to separation at lower temperature gradients.

## Enrichment in geological networks

The above measurements were performed in a single heat flow chamber, however, geological systems of interconnected fractures are highly variable in size and can span from millimetres to tens or even hundreds of metres. To approach such complex systems, we first performed a proof-of-principle experiment (Fig. 4a−c). We applied a flow of 1 nl s⁻¹ of mixed compounds to the first heat flux chamber 1 (violet shade, Fig. 4a), which branches into chambers 2 and 3 (brown and blue shades, Fig. 4a). To determine the spatial enrichment, we ran a mixture of amino acids through the chambers ($\Delta T = 16$ K) for 60 h, froze them and divided them into the colour-graded areas shown in Fig. 4a for analysis through HPLC. Figure 4b shows the position and concentration of exemplarily selected amino acids phenylalanine (F), I and asparagine (N) with the corresponding colour assignment (Extended Data Fig. 6 and Supplementary Fig. 4 for various amino acids). The scatter plots, including I and N, show the separation of concentration ratios per chamber. The range of enrichments in a three-chamber network ([N]/[I] = 23-fold and [I]/[N] = 4.2-fold) is higher than achievable in a single chamber (Fig. 4b, left and Supplementary Fig. 5). This shows that the enrichment effect

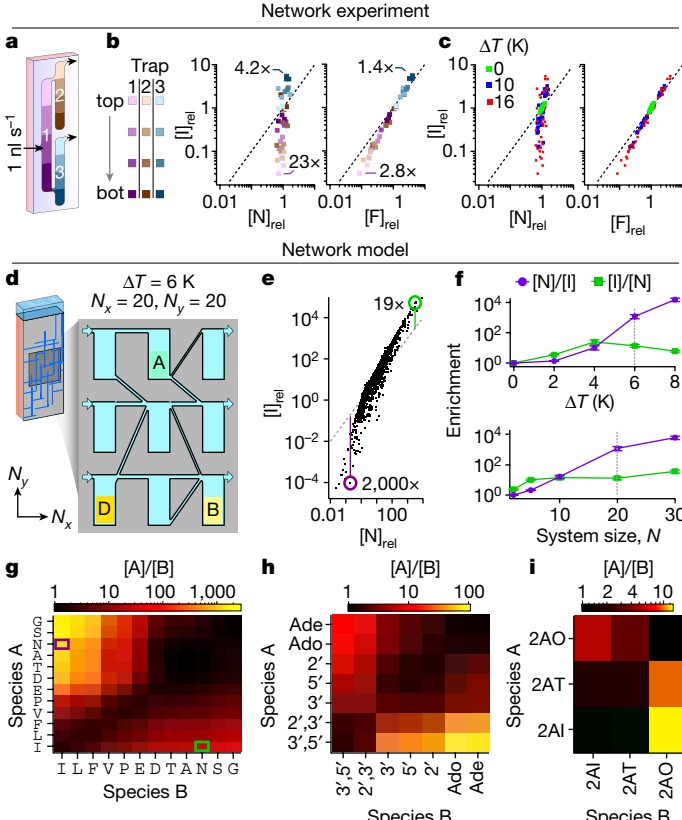

**Fig. 4 | Experimental and modelled purification of prebiotic organics in a network of connected rock cracks. a**, Experimental setup of a small network of three interconnected chambers with a volume inflow of 1 nl s⁻¹ of an amino acid mixture and $\Delta T = 16$ K. **b**, After 60 h, the chamber contents from three repeats were frozen and divided into individual parts according to the colour gradations in **a** and measured by HPLC. Exemplary separations of amino acids I versus N and I versus F are shown. Concentration ratios in chamber 2 (brown shade) versus 3 (blue shade) range from [I]/[N] = 23-fold (brown) to [N]/[I] = 4.2-fold (blue). For thermophoretically similar amino acids F and I, these range from [I]/[F] = 2.8-fold (brown) to [F]/[I] = 1.4-fold (blue). The dashed black line indicates equal concentration. **c**, Data from **b** in red compared with an otherwise identical run at $\Delta T = 10$ K (blue) and 0 K (control, green) show stronger enrichment at higher temperatures and no effect without heat flux.

**d**, Larger systems ($N_x \times N_y = 20 \times 20$, $\Delta T = 6$ K) of connected heat flow chambers with a volume inflow of 1 nl s⁻¹ per input channel are modelled numerically. **e**, Concentration ratios between amino acids I and N in the chamber system show a similar but amplified pattern as in **b**. **f**, Maximum enrichments $\overline{[N]/[I]}$ (purple) and $\overline{[I]/[N]}$ (green) scale with the temperature difference and the size of the system. The dashed lines show the conditions in **e** and **g**. Error bars represent the s.d. from several simulations (Methods). **g**–**i**, Maximum enrichments in the system shown in **d** for all possible combinations of substances for a mixture of amino acids (**g**), adenine (nucleobase/nucleoside/nucleotides) (**h**) and 2-aminoazoles (**i**). Highlighted boxes indicate maximum enrichments for the pair I and N shown in **e**. See Extended Data Fig. 6 for error values.

is compatible with throughflows and is boosted substantially by the interaction of three chambers in a network. Maximum enrichments for thermophoretically different amino acids, such as I versus N, are much greater than those for thermophoretically similar amino acids, such as I versus F (Fig. 4b, right and more examples in Extended Data Fig. 6). As expected, the enhancement becomes stronger for larger temperature differences (Fig. 4c, blue $\Delta T = 10$ K, red $\Delta T = 16$ K) and disappears without a heat flux (green, $\Delta T = 0$ K).

These results encouraged us to model the behaviour of larger systems in silico. We numerically assembled a network of $N_x = 20$ by $N_y = 20$ connected heat flow chambers with a size of 50 × 200 mm, fed by a geothermal fluid flow (Fig. 4d). To understand the behaviour of each species inside the complete system, we first determined the thermophoretic properties of all solutes (Supplementary Code and Data and Extended Data Table 1; Methods). For 3′,5′-AMP, one of the few compounds used in this work with known thermophoretic properties, our result $S_{T,3',5'\text{-AMP}} = (0.0051 \pm 0.0009)$ K⁻¹ (at 40 °C) agrees with the literature values $S_{T,3',5'\text{-AMP,ref}} = 0.005$ K⁻¹ (at 30 °C) within the error margin[7].

On the basis of these results, we calculated the average concentrations for mixtures of prebiotic compounds in the lower parts of all network chambers for 30 separate networks and a temperature difference

of 6 K (Fig. 4e–i). The flow rate for the inlet channels, which mimic the slow geothermal flow, was set to 1 nl s⁻¹ as assumed realistic in geological systems[49] and used in the experiments in Fig. 4a–c. The flow rates for the individual chambers of the network mostly varied between 0.1 and 10 nl s⁻¹ as a result of the randomly assigned throughputs of the connecting channels (Supplementary Fig. 10).

Similar to the experimental results shown in Fig. 4b,c, species with high Soret coefficients are strongly concentrated in upstream chambers and depleted downstream. There, species with small Soret coefficients are dominant and, therefore, highly enriched (Extended Data Fig. 9). Accordingly, the maximum ratio of concentrations of the amino acids I versus N is 19-fold at higher absolute concentrations, whereas the ratio of N versus I in another chamber is 2,000-fold at reduced absolute concentrations (Fig. 4e). Each point in Fig. 4e represents the average concentrations in the lower part of the respective heat flow chamber of the network, showing a similar but amplified pattern compared with the experiments. Extended Data Fig. 7 shows further examples of amino acid combinations analogous to Fig. 4e, demonstrating enrichments at notable absolute concentrations.

In natural systems, temperature differences across chamber cross-sections are eventually below the assumed $\Delta T = 6$ K. For this

reason, we varied the temperature difference in Fig. 4f and found a near-exponential dependence for the enrichment of N versus I. The reverse enrichment, that is, I versus N, saturates at $(25 \pm 9)$-fold and even declines at higher temperature differences (Fig. 4f, top), owing to the throughflow-limited thermophoretic accumulation of solutes at the chamber bottom at higher temperature gradients. By contrast, in the upper chamber region, species with strong thermophoresis can deplete almost completely, leading to high enrichments of species with smaller Soret coefficients with increasing temperature gradients (Supplementary Fig. 11). The errors were determined from three repeated simulations, considering the errors of the Soret coefficients (Extended Data Table 1; Methods). Notably, even a slight temperature difference of only 2 K drives an up to $(3.5 \pm 0.2)$-fold enrichment of I versus N (Supplementary Discussion 1). The enrichment is further boosted with increasing network size, in particular for the thermophoretic weaker species (N versus I, Fig. 4f, bottom).

Figure 4g shows the maximum enrichment of amino acids under the same model parameters as in Fig. 4e. The network amplifies the respective values compared with Fig. 2b; for example, G is enriched up to $(2,726 \pm 713)$-fold compared with I; in other chambers, I is enriched $(38 \pm 14)$-fold. Amino acids with similar Soret coefficients, such as I and V, are enriched up to a maximum of $\overline{[I]}/\overline{[V]} = (1.6 \pm 0.1)$-fold. Similar effects were observed between nucleobases and nucleotides of different phosphorylation states and 2-aminoazoles (Fig. 4h–i and Extended Data Fig. 6).

Starting with a 1:1 solution of two substances with sufficiently different thermophoresis, such as I and N, heat fluxes in networks can thus provide niches with at least 95% purity of the stronger accumulated substance. For the thermophoretically weaker compound, even 99.9% purity is feasible.

## Habitat for prebiotic reactions

How does prebiotic chemistry benefit from the heat-flow-driven purification outlined above? We address this question using the example of a TMP-driven reaction. As a highly water-soluble phosphate species, TMP is particularly interesting for prebiotic chemistry, enabling various prebiotic reactions even in water or pasteous environments[13,19]. However, TMP is considered scarce on the prebiotic Earth because of energy-intensive synthesis pathways, making its selective enrichment critical[14].

As an example of a TMP-driven reaction, we examine the dimerization of glycine in water[11,12] (Fig. 5a). We filled a heat flux chamber with a mixture of 1 mM (Fig. 5b, pink) or 10 mM glycine (Fig. 5b, purple) and 1 mM TMP. After 16 h runtime with a temperature difference of $\Delta T = 14$ K, the product yields were increased from an undetectable level ($\Delta T = 0$ K, Fig. 5b, black, $T = 85$ °C) to $(3.6 \pm 0.6)$%, enabled by the selective enrichment of TMP over the thermophoretically weaker glycine (Extended Data Table 1).

Encouraged by these results, we sought to model the dimerization of glycine in a network of heat flow chambers. We first determined the reaction rates of the reaction model (equations (9)–(13)), studying the dimerization of glycine experimentally in bulk under TMP titration at 90 °C and with initial pH 10.5 over up to 120 h (Fig. 5c; Methods). The model network consists of $20 \times 20$ connected heat flow chambers, fed with a mixture of TMP and glycine (1 μM each) at a flow rate of 1 nl s⁻¹ per inlet (Fig. 5d).

In the absence of heat flows, the concentrations of the product GlyGly are vanishingly small at around 10 fM (Fig. 5e, solid lines). The situation changes substantially on applying a temperature difference of 10 K to each chamber. The heat-flow-driven enrichment increases the maximum reactant concentrations by four orders of magnitude (blue and black dashed lines, Fig. 5e) and enhances the maximal product yields $2c_{GlyGly}/(2c_{GlyGly} + c_{Gly})$ by up to five orders of magnitude, reaching around 10% at $\Delta T = 10$ K (Fig. 5f). These results

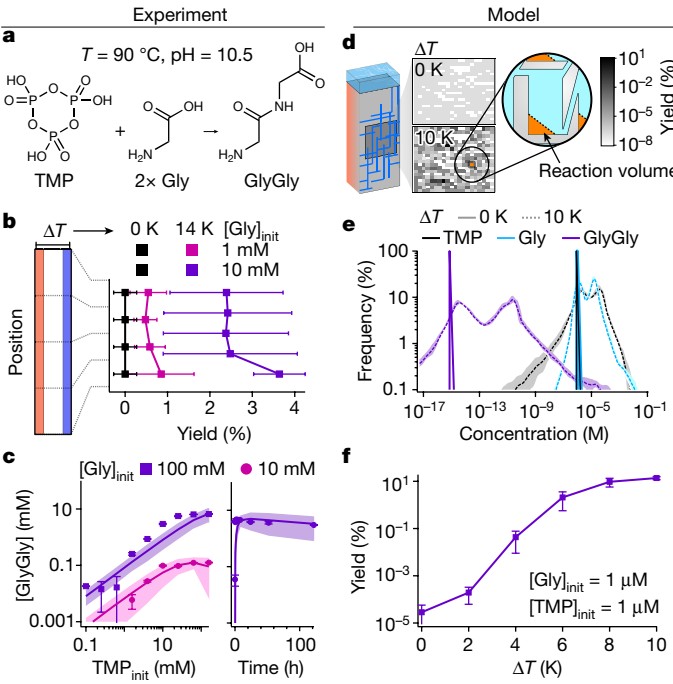

**Fig. 5 | Experimental and modelled enhancement of reaction yields by heat-flow-driven, selective purification of reactants. a**, TMP-associated dimerization of glycine[11]. **b**, Even in a single heat flux chamber ($\Delta T = 14$ K), the product yield of Gly dimerization increases to up to $(3.6 \pm 0.6)$% after 16 h from undetectable yields in the controls ($\Delta T = 0$ K) for $[Gly]_{init} = 1$ mM and 10 mM and $[TMP]_{init} = 1$ mM (errors = s.d., three repeats). **c**, The reaction rates from equations (9)–(13) were determined experimentally by simultaneously fitting to the product concentrations for bulk glycine dimerization after 16 h with 10 mM and 100 mM initial Gly concentrations over three orders of magnitude of TMP concentrations (left) and to a time series up to 120 h for $[Gly]_{init}$ and $[TMP]_{init} = 100$ mM (right). **d**, Gly dimerization was modelled numerically in networks of connected cracks by using reaction rates obtained in **c**. The 2D maps show product concentrations of GlyGly from TMP-induced dimerization after 120 h without ($\Delta T = 0$ K) and with ($\Delta T = 10$ K) applied heat flux with an inflow rate of 1 nl s⁻¹ per input channel. Each pixel corresponds to the bottom concentration of GlyGly in a separate crack (reaction volume, orange). **e**, Statistics of all reactant concentrations in 100 distinct systems of $N_x = 20$ by $N_y = 20$ connected cracks. Without heat flow ($\Delta T = 0$ K, solid line), TMP (black) and Gly (blue) concentrations remain unchanged at 1 μM, leading to vanishingly low product concentration (GlyGly, purple). With heat flow ($\Delta T = 10$ K), reactant concentrations are increased more than four orders of magnitude, so that—in 0.1% of all chambers—at least 10 μM of product can be formed. Shaded area denotes the s.d. by several runs (Methods). **f**, The reaction yields of TMP-driven glycine dimerization increase exponentially with the applied temperature gradient.

show that otherwise challenging prebiotic reactions are massively boosted by the heat-flow-driven selective accumulation and local enrichment.

## Conclusion

In summary, we have studied experimentally and numerically the selective and localized enrichment of more than 50 prebiotically relevant substances by heat fluxes. The scenario mimics the situation in extended geological networks of rock cracks and shows how geothermal heat flows could have driven the sensitive separation of highly similar molecules. At moderate temperature differences, distinct amino acids are separated from each other by up to three orders of magnitude, components of nucleotides by up to two orders of magnitude and the different 2-aminoazoles by a factor of 10. In the upstream part of the

network, species with strong thermophoresis are accumulated and depleted further downstream. Here residual chemicals with weak thermophoresis are enriched at moderate thermophoretic accumulations (Extended Data Figs. 6, 7 and 9). The resulting concentration ratios could be conveyed to higher absolute concentrations by non-selective up-concentration modes such as local drying or accumulation at gas–water interfaces. As long as the thermophoretic strength of two substances differs sufficiently, they can be spatially separated, enriched and mixed with other concentrated chemicals in the geo-microfluidic system.

The mechanism works over a wide range of pH (3–11) and solvent conditions (Extended Data Figs. 3–5). The strength of the enrichment scales exponentially with the temperature difference. Smaller heat fluxes could have been compensated by naturally occurring larger, metre-scaled networks. The process is very stable to local and temporal fluctuations in the heat flux and operates in a variety of irregular chamber geometries (Supplementary Discussion 1 and Supplementary Fig. 12). All investigated compounds exhibit considerable thermophoretic accumulation that scales exponentially with their Soret coefficients, the latter spanning a wide range between $1.4 \times 10^{-3}$ and $7.5 \times 10^{-3}\ K^{-1}$.

We demonstrate the advantages for prebiotic chemistry by placing an otherwise challenging reaction, glycine dimerization, in a network of interconnected cracks. Such reactions typically suffer from the low relative abundance of their driving agents, in our case, TMP. We show experimentally that the thermal non-equilibrium markedly enhances reaction yields in a single chamber and numerically demonstrate how this effect is amplified by the network, boosting yields by up to five orders of magnitude. The universal availability of heat fluxes on the early Earth, either from the geological setting or as a waste product of one of many exothermic reactions, makes this mechanism conceivable in various environments.

Systems of interconnected thin fractures and cracks or comparable permeable pathways are thought to be ubiquitous in volcanic and geothermal environments. Connected to the surface, such systems can potentially feed spatially separated ponds or pools, whose role in the origin of life has been extensively studied. Ultimately, a large number of sequential reaction conditions required by numerous prebiotic reaction pathways could have been implemented without external intervention. Given the wide availability of heat flows and fractures in rocks, the observed applicability to even small prebiotic compounds and the overall robustness of the process, thermophoretic enrichment of organics could have provided a steady driving force for a natural origins-of-life laboratory.

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

# Methods

## List of substances used

For a list of abbreviations, see Supplementary Table 1. Nucleobases A and T, nucleosides Ado, Guo and Urd, nucleotides (2′-AMP, 3′-AMP, 5′-AMP, 2′,3′-AMP, 3′,5′-AMP, 5′-CMP, 2′,3′-CMP, 5′-GMP, 3′,5′-GMP and 5′-UMP), deoxyribonucleotides (dAMP, dCMP, dGMP and dTMP), 2-aminoazoles (2AO, 2AI and 2AT), oligomers of Gly (Gly, GlyGly and GlyGlyGly) and TMP were purchased from Sigma-Aldrich (USA). Nucleobase C was purchased from Carl Roth GmbH (Germany) and nucleobases U and G and nucleosides L-Cyd and D-Cyd were purchased from Biosynth Carbosynth (UK). Nucleotides 2′-CMP, 3′-CMP, 3′,5′-CMP, 2′,3′-GMP, 2′,3′-UMP and 3′,5′-UMP were purchased from Biolog (USA). For the proteogenic amino acids, an Amino Acid Mixture was used (L4461, Promega, USA) and compared with individual amino acids (Sigma-Aldrich, USA). For experiments on non-proteogenic amino acids, a mixture of amino acids and small molecules was used (A9906, Sigma-Aldrich, USA).

## Experimental procedure: preparation of heat flow cells

Bold numbers refer to the corresponding encircled numbers in Extended Data Fig. 1. The FEP foils (**5**; Holscot, the Netherlands) defining the microfluidic structure were cut by an industrial plotter device (CE6000-40 Plus, Graphtec, Germany). The cutout was then sandwiched between two sapphires (**4** and **6**; KYBURZ Switzerland, Switzerland) with thicknesses of 500 μm (cooled sapphire) and 2,000 μm (heated sapphire), respectively. The sapphires were previously coated with a hydrophobic coating (ProSurf MT-5, Surfactis, France) to avoid interaction with the sample and facilitate sample extraction. The cooled sapphire (**4**) has four laser-cut holes with a diameter of 1 mm. The sapphire–FEP–sapphire block was then placed on an aluminium base (**2**), covered by a heat-conducting foil (**3**; graphite, 25 μm, 1,600 W mK$^{-1}$, Panasonic, Japan) and held in place by a steel frame (**7**), which is connected to the aluminium base by six torque-controlled steel screws for homogeneous force distribution.

The height of the chamber was measured with a confocal micrometer (CL-3000 series with CL-P015, Keyence) at three positions (bottom, middle and top) to ensure a homogeneous thickness of the chamber. The average of these three measurements is later used in the numerical model to determine the Soret coefficient for each experiment. An Ohmic heating element (**9**) was mounted on the steel frame with torque-controlled screws and connected to the heated sapphire with another heat-conducting foil (**8**; EYGS0811ZLGH, graphite, 200 μm, 400 W mK$^{-1}$, Panasonic).

For the microfluidic connections, we used (Techlab, Germany): connectors (UP P-702-01), end caps (UP P-755), screws (VBM 100.823-100.828), ferrules (VBM 100.632) and tubings (Teflon (FEP), KAP 100.969). As syringes, we used (ILS, Germany, bought from Göhler-HPLC Syringes, Germany) 2606814 and 2606035. Chambers were pre-flushed using low-viscosity, fluorinated oil (3M Novec 7500 Engineered Fluid, USA) to check for tightness and push out residual gas inclusions.

For the 100 mM phosphate buffer used in Extended Data Fig. 5c, we dissolved 584 mg NaH$_2$PO$_4$·H$_2$O and 819 mg Na$_2$HPO$_4$ in 100 ml water, resulting in a solution of pH 7. Each mixture was chosen to cover a broad range of molecules while still producing separable peaks in the HPLC measurements (see Extended Data Fig. 1 and Supplementary Fig. 14).

The sample (42.5 μl) was loaded into the chamber with the help of a syringe filled with fluorinated oil to avoid the inclusion of air bubbles. After loading the sample, the tubings were closed with end caps.

The chamber was then mounted to a cooled aluminium block connected to a cryostat (TXF-200 R5, Grant, UK). The heaters were connected to a 400-W, 24-V power supply controlled by means of Arduino boards with a customized version of the open-source firmware Repetier, initially designed for 3D printing. The cryostat and the heaters were set to the desired temperatures. Temperatures were measured on the sapphires with a heat imaging camera (ShotPRO, Seek Thermal, USA), giving cold and hot temperatures per experiment.

All elements were optimized using finite element simulation to provide a homogeneous temperature profile.

To stop the experiment, heaters and the cryostat were turned off and the chamber was frozen at −80 °C for at least 15 min. This allowed us to open the chamber and cut the frozen interior into four fractions of equal volume. Fewer fractions would lower resolution by averaging over a larger fraction of the chamber (Extended Data Figs. 1 and 2), whereas more fractions would suffer a larger positional error from manual cutting (for a comparison of 12 versus four fractions, see Extended Data Fig. 2).

For experimental network experiments, a modified cutout was used (shown in Fig. 4). Before mixing with concentrated amino acid standard, the water used for dilution was degassed by heated stirring under vacuum. The sandwich was filled with sample solution before assembly to minimize air inclusions and dilution effects from remaining water. To start the experiment, inlets and outlets were connected to tubings (250 μm inner diameter to reduce air inclusions, KAP 100.966, Techlab, Germany) filled with sample solution and connected to syringes driven by syringe pumps (Nemesys, CETONI, Germany). As soon as the temperature gradient (16 K between 30 °C and 46 °C and 10 K between 32 °C and 42 °C) was established, the inlet was supplied with a continuous flow of 1 nl s$^{-1}$ and the outlets with 0.5 nl s$^{-1}$ each. For the recovery of fractions of the three individual chambers, a slightly modified extraction procedure was used. Because the time required to extract 12 fractions (four fractions per chamber) is increased compared with the four fractions used above, we added dry ice around the frozen block to reduce dilution by condensation. Also, we directly pipetted the recovered fraction into 7 eq. of borate buffer for pre-column derivatization.

## pH measurements

For pH measurement and adjustment, we used a Versa Star Pro device equipped with an 8220BNWP micro pH electrode (Thermo Fisher Scientific, USA). The fundamental effect of heat-flow-driven pH gradients was shown previously[41]. However, given the low concentrations of solutes used in this work, the obtained pH differences remained weak (delta pH between 0 and 1 units).

## LC + MS measurements

Two different LC systems were used for the various analysis methods for all molecules of interest. Also, for glycine dimerization and detection of non-proteogenic amino acids, an orbitrap MS was used. A method was created and optimized for each set of species to ensure the best separation. In the following, these will be listed for the molecules involved.

System 1: the system consists of a Vanquish Flex (VF-S01-A), a binary pump (VF-P10-A-01), a heated column compartment (VH-C10-A) and a variable-wavelength detector (VF-D40-A; all Thermo Fisher Scientific, USA). As column, we used Symmetry C18 (3.5 μm pore size, 2.1 mm diameter, 150 mm length, 100 Å particle size, WAT106005) (Waters, USA). As eluents: eluent A: LC-H$_2$O (0.1% v/v formic acid); eluent B: LC-acetonitrile (0.1% v/v formic acid). For all methods, we used a flow of 0.3 ml min$^{-1}$, a column temperature of 30 °C (still air) and detection of the UV absorption at 260 nm with 50 Hz. All methods are followed by a washing step at 40% B for 1 min and equilibration at starting concentration for 6 min.

Detailed protocols for the respective mixtures of compounds:
- Nucleobases: isocratic elution for 3 min at 0% B, then increase to 10% B over 2 min.
- Nucleosides: isocratic elution for 3 min at 0% B, then increase to 5% B over 4 min.
- Cytidine monophosphates: isocratic elution for 5 min at 0% B, then increase to 5% B over 2 min.
- Adenosine monophosphates: start with 0% B, then increase to 5% B over 12 min.

- 5′-ribonucleotides: isocratic elution for 10 min at 0% B.
- 2′,3′-cyclic ribonucleotides: start with 0% B, then increase to 4.5% B over 15 min.
- 3′,5′-cyclic ribonucleotides: start with 0% B, then increase to 6.8% B over 9.5 min.
- Deoxyribonucleotides: start with 0% B, then increase to 3% B over 8 min, then increase to 5% B over 4 min and 7.5% B over 1 min.
- Dimerization of glycine (variation of TMP): pre-column derivatization as described below. Isocratic elution for 10 min at 4% B.

MS-based methods: we carried out mass analysis using a Q Exactive Plus Orbitrap HR/AM (Thermo Fisher Scientific, USA), using positive ionization with a resolution of 70k, an AQC target of $3 \times 10^6$ and a maximum IT of 200 ms. On the HESI source, a sheath gas flow rate of 2 was set, a spray voltage of 2.9 kV, 320 °C capillary temperature, 50 °C auxiliary gas heater temperature and a S-lens RF level of 50. For analysis, the main isotope mass ±0.075 $m/z$ was extracted. For LC-MS methods, we used the same settings for the LC as above.

Detailed protocols for the respective mixtures of compounds:
- Dimerization of glycine (time variation and heat flow chambers): isocratic elution for 5 min at 4% B.
- Mixture of non-proteogenic amino acids, proteogenic amino acids, dipeptides and other molecules: isocratic elution for 5 min at 4% B.

System 2: the system consists of a Vanquish Core (VC-S01-A-02), a quaternary pump (VC-P20-A-01), a heated column compartment (VC-C10-A-03), a diode array detector (VC-D11-A-01) and a fluorescence detector (VC-D50-A-01; all Thermo Fisher Scientific, USA).

Separation of 2AO, 2AI and 2-aminothiazole (2AT): following a method from the literature[50], we applied isocratic elution for 5 min with 90% LC-H$_2$O, 10 mM ammonium formate and 10% LC-acetonitrile. The flow rate of 1.5 ml min$^{-1}$ was applied on the column InertSustain Amide (100 Å, 5 µm, 4.6 × 150 mm, GL5020-88631) (GL Sciences, Japan), with a column temperature of 40 °C and UV detection at 225 nm.

Separation of all 20 proteogenic amino acids: for the separation of amino acids, we adapted a method from the literature[51]. Following that protocol, we first pre-column derivatized our sample. This was done by mixing 14 µl of 50 mM borate buffer (28341, Thermo Fisher Scientific, USA) with pH adjusted to 8.8 and 2 µl of our sample. Then, 4 µl of freshly prepared 6-aminoquinolyl-N-hydroxysuccinimidyl carbamate (AQC, S041, Synchem, USA) and 4 mg ml$^{-1}$ in anhydrous acetonitrile (43166, Alfa Aesar, USA) were added and mixed with a pipette. The samples were then incubated for 10 min at 55 °C and stored in the autosampler at 5 °C before injection. We used the column ACCLAIM Vanquish C18 (2.2 µm, 2.1 mm × 150 mm) (Thermo Fisher Scientific, USA). Eluent A was LC-H$_2$O (W6-212, Fisher Scientific, USA) + 50 mM ammonium formate (17843, Honeywell, USA) + 0.8% v/v formic acid (A117-50, Fisher Scientific, USA) and eluent B was LC-acetonitrile (A955-212, Fisher Scientific, USA). We set the column temperature at 45 °C (still air) and simultaneously acquired the UV absorption signal at 260 nm and the fluorescence signal (FLD) under excitation at 266 nm and emission at 473 nm (both with 50 Hz). Elution was done with a flow of 0.65 ml min$^{-1}$. Best separation was achieved using the following elution protocol: isocratic elution with 0.5% B between 0 and 0.548 min, increase to 5.2% B until 3 min, to 9.2% B until 8.077 min, to 14% B until 8.626 min, maintain at 14% B until 9.5 min, increase to 19.2% B until 11.227 min, to 19.5% B until 13.696 min, to 90% B until 14.4 min and finally lower to 0.5% B and equilibrate for 6 min. For peak identification, samples of the individual amino acids were prepared according to the same protocol. Calibrations with different concentrations of amino acid standard solutions were done using the same volume as for experimental samples and for each set of measurements. After a comparison of the results of these calibrations in UV and FLD measurements, we found that the two detection methods worked equally. Because background levels in FLD are very low, we chose to proceed with the FLD channel

except for tryptophan, which does not fluoresce, and tyrosine, which is known to produce intermediary products[51]. For Fig. 2 and Extended Data Fig. 3, we injected each sample twice and used the average of these two injections.

**Treatment of data**

Integral values of all experiments were analysed using Python. The accumulation profiles (Fig. 2a and Extended Data Figs. 2a and 8) over the chamber height were internally normalized for each species (see equation (1)) to compare enrichments despite the different initial concentrations owing to the use of different mixtures. Inherently, this makes the system compatible with mixtures with different concentrations of species. The normalized concentration [A]$_{j,k}$ for each individual measurement was calculated for species A for fraction $j \in \{1, 2, 3, 4\}$ and replicate number $k \in \{1, 2, 3\}$.

$$[A]_{j,k} = \frac{[A]_{j,k,\text{HPLC}}}{\frac{1}{4} \sum_j [A]_{j,k,\text{HPLC}}} \tag{1}$$

with the HPLC-measured concentration [A]$_{j,k,\text{HPLC}}$ obtained from the integral values presented in Supplementary Tables 5–65. For the detection of the dimerization of glycine, we used a quadratic function of ($\sqrt{\text{response}}$) to account for the nonlinear response (Supplementary Information pages 166–167). For all other species, we used an external linear calibration (Supplementary Tables 3 and 4 and examples on pages 135–165 of the Supplementary Information). The linear calibration intrinsically emphasizes that differences in calibration or absolute concentration do not change the resulting enrichments.

To compare two species inside a pool of molecules (as shown in the heat maps in Figs. 2 and 3 and Extended Data Figs. 2–5), we calculated the concentration ratio of species A compared with species B in the top ($j = 1$) and respective bottom ($j = 4$) fraction for each combination of species and per individual experiment. We then calculated the average of the triplicate experiments.

$$\overline{[A]_j/[B]_j} - 1 = \frac{1}{3} \sum_k \left( \frac{[A]_{j,k}}{[B]_{j,k}} \right) - 1 \tag{2}$$

The averaging over the triplicates is done only after the calculation of the concentration ratios. This is necessary because the temperature gradients between the replicate experiments differ slightly (1–2 K). This affects the concentrations of all species present in the respective mixture of a replicate equally (Supplementary Fig. 9), so that a calculation of the concentration ratio only after averaging the species concentrations would lead to a distortion of the enrichment value actually present in the heat flow chamber. The error maps shown in the Extended Data Figures and Supplementary Figures were determined by calculating the s.d. of the enrichments [A]$_{j,k}$/[B]$_{j,k}$ of all repeats from the average value described in equation (2).

Control experiments were done in parallel to check for possible degradation of individual compounds. For this purpose, one solution each was incubated at the lower or higher temperature occurring in the heat flow chamber in bulk for the duration of the accumulation experiments, but no substantial selective degradation was observed.

For the enrichment of an individual species compared with a pool of molecules (Supplementary Table 2), we normalized the individual experiments as explained previously. Then, we calculated the average concentration $\overline{c}_{j,k}$ of the pool of all species $k$ (per experiment) in the top ($j = $ top) and respective bottom ($j = $ bot) fraction for species $i \in \{1, ..., S\}$.

$$\overline{c}_{j,k} = \frac{1}{S} \sum_i \frac{c_{i,j,k}}{\frac{1}{4} \sum_j c_{i,j,k}} \tag{3}$$

By comparing the concentration of the species with the mean concentration of the pool $\overline{c}_j$, we were able to determine the enrichment of

species A in fraction $j$ (per experiment), which we averaged over the triplicate measurements (Supplementary Table 2).

$$\overline{[A]}_j / \overline{c}_j - 1 = \frac{1}{3} \sum_k ([A]_{j,k} - \overline{c}_{j,k}) - 1 \qquad (4)$$

## Determination of the thermophoretic strength (Soret coefficient) of prebiotic substances in complex mixtures

A new measurement approach was necessary to determine the thermophoretic properties of the very small and highly diluted prebiotically relevant chemicals mixed together in networks of interconnected rock fractures. Previous methods require either fluorescent labelling of the measured substance, which distorts the thermophoretic properties of small molecules, or high substance concentrations, which—in our example—would strongly change the pH value and, thus, would not be representative for the prebiotic context of diluted solutions. At the same time, only one substance can be measured at a time. To overcome these difficulties, we use the same setup that mimics the geological scenario of heat flows through thin rock fractures that has also been used in previous studies[40,42] (Extended Data Fig. 1). The microfluidic structure is prepared as explained earlier. The combination of thermal convection and thermophoresis allows the solutes to be separated along its height of 50 mm much more effectively than would be possible by thermophoresis alone[52]. This spatial separation allows the entire chamber to be frozen. The content of the chamber is then cut into four equal-sized pieces (12.5 mm height each) using a scalpel. All pieces are analysed separately by HPLC (Extended Data Fig. 1), resulting in an average concentration $\overline{[A]}_j = 1/3 \sum_k [A]_{j,k}$ of species A at position $j$, with $[A]_{j,k}$ as introduced in equation (1).

We assumed diffusive mobilities of $D_{i,AA} \approx 800 \ \mu m^2 \ s^{-1}$ for amino acids[53] and $D_{i,AA} \approx 1,400 \ \mu m^2 \ s^{-1}$ for the 2-aminoazoles and nucleotide components[54]. This approximation is reasonable because the experiment is close enough to steady state after 18 h and, hence, the fitted thermophoretic strength $S_{T,i}$ depends only marginally on $D_i$ (Supplementary Fig. 8).

To determine the thermophoretic strength from these datasets, we first create a 2D model of the same height (50 mm) and thickness (0.17 mm) as in the experiment using finite element methods (COMSOL 5.4). In this model, the temperatures of the sidewalls are set according to the experiment. To determine the thermal convection of the solvent, we solve the Navier–Stokes equation

$$\rho(\mathbf{u} \cdot \nabla)\mathbf{u} = \nabla \cdot \left[ -p\mathbf{I} + \mu(\nabla\mathbf{u} + (\nabla\mathbf{u})^T) - \frac{2}{3}\mu(\nabla \cdot \mathbf{u})\mathbf{I} \right] + \rho\mathbf{g} \qquad (5)$$

and the continuity equation in the steady-state case

$$\nabla \cdot (\rho\mathbf{u}) = 0 \qquad (6)$$

in which $\rho$ denotes solvent density, $\mathbf{u}$ the solvent velocity vector, $\mathbf{I}$ the unit vector, $p$ the pressure, $\mu$ the dynamic viscosity and $\mathbf{g}$ the gravitational acceleration. The results show a laminar convection flow $\mathbf{u}$ inside the cell, which is coupled to the solute movement as it drags it with it.

This is achieved by solving the time-dependent drift–diffusion equation

$$\frac{\delta c_{i,num}}{\delta t} = \nabla \cdot [D_i \nabla c_{i,num} - (\mathbf{u} - S_{T,i} \cdot D_i \nabla T) c_{i,num}] \qquad (7)$$

with the local solute concentration $c_{i,num}$ of species $i$ (initial concentration $c_{i,num,0} = 1$), its diffusive mobility $D_i$ and Soret coefficient $S_{T,i} \equiv \frac{D_{T,i}}{D_i}$, which is defined using its thermophoretic mobility $D_{T,i} = -\frac{v_{T,i}}{\nabla T}$, including the thermophoretic drift velocity $v_{T,i}$. $D_i$ is determined approximately by literature values after making sure that it does not change the determination of $S_{T,i}$ (Supplementary Fig. 8). To obtain $S_{T,i}$ for each

species of the mixture, we fitted the numerical results of the averaged concentrations of all four volume fractions $V_j$, $\overline{c}_{i,j,num} = \int_{V_j} c_{i,num} dV$ to the average concentrations obtained by HPLC $\overline{[A]}_j$, $i = A$. Although external control of COMSOL is possible to include it directly in the fitting algorithm, we chose to first solve equations (5)–(7) for a wide range of parameters and then use this dataset to linearly interpolate the experimental results, yielding the respective Soret coefficient of the solute much faster and with good precision. The parameter range covered different temperature gradients ($\Delta T$ [K]: 0, 5, 10, 20, 25), thermophoretic mobilities ($D_T$ [$\times 10^{-12} \ m^2 \ sK^{-1}$] : 1, 2.5, 5, 10, 20, 40, 60, 80) and diffusive mobilities ($D$ [$\times 10^{-12} \ m^2 \ s^{-1}$] : 1, 700, 1, 400, 1, 100, 700) for 100 equidistant time points between 0 h and 24 h, which results in 192 different concentration profiles at each time point (see model file SimpleSim_2022_03_08_new.mph, resulting in the data file 2022_03_08_2DSim.dat). Using a custom-made LabVIEW program incorporating Levenberg–Marquardt algorithms (see SingleNTD_TrapFitter_V1-6. llb), we varied the $D_{T,i}$ with fixed, experimentally obtained values for $\Delta T$, $D_i$ until an optimal fit between the numerical ($\overline{c}_{i,j,num}$) and experimental ($\overline{[A]}_i$) concentration profiles was found for each species. This procedure was repeated for all experimental repeats (triplets), after which the obtained values for $S_{T,i}$ were averaged and a s.d. was obtained (Extended Data Table 1). Steady-state concentration profiles shown in Extended Data Fig. 2c,d and Supplementary Fig. 6 were obtained by calculating

$$c_i(y) = \exp\left( -\frac{\frac{q_i}{120}}{1 + \frac{q_i^2}{10,080}} S_{T,i} \Delta T \frac{y}{\alpha} \right), \ \text{with} \ q \equiv \frac{\Delta T \beta g \rho \alpha^3}{6\eta D_i} \qquad (8)$$

in which $\beta$ denotes the volume expansion coefficient of water, $\alpha$ the distance between the hot and the cold sides of the heat flow chamber, $\eta$ the dynamic viscosity of water and $y$ the space coordinate along the height of the chamber.

## Modelling of a system of connected cracks/heat flow chambers

To determine the concentration profiles in a system of interconnected heat flow chambers from the previously determined Soret coefficients, we first had to calculate the behaviour of all species in a single heat flow chamber under various boundary conditions, such as temperature difference and flow rates.

For this, we created a 3D chamber in COMSOL with a height of 200 mm, a width of 60 mm and a thickness of 0.17 mm. The chamber has an inflow channel at its top end and an outflow channel at its top and bottom ends (see supplied COMSOL file SingleNTD_v7_simpleGeo_Large_allSTs_60mmWide200mmHighTrap.mph, yielding data file SingleNTD_v7_simpleGeo_Large_60x200mmWide.dat). We then solve equations (5)–(7) in the steady-state case, calculating first the laminar flow and then the concentration distribution of the solute. Solutions are generated for all combinations of boundary conditions, that is, for different temperature differences ($\Delta T$ [K]: 0, 1, 2, 3, 4, 5, 6, 7, 8, 9, 10, 11, 12), diffusive mobilities ($D_i$ [$\times 10^{-12} \ m^2 \ s^{-1}$] : 800, 1, 400), Soret coefficients ($S_{T,i}$ [$\times 10^{-3} K^{-1}$] : 1, 4, 8, 12) covering the range given by Extended Data Table 1, inflow volume rates ($Q_{in}$ [$\times 10^{-2} \ nl \ s^{-1}$] : 1, 5, 10, 50, 100, 500, 1, 000) and outflow rates through the bottom outflow channels as ratios of the inflow volume rate ($Q_{out,bot}$ [%]: 1, 5, 10, 20, 50, 100). Because the random distribution of channels in the modelled network of heat flow chambers results in channels with flow rates smaller and larger than the assumed 1 nl s$^{-1}$ (Supplementary Fig. 10), we appropriately set the range of simulated flow rates $Q_{in}$. The outflow rate of the upper channel is determined from the mass conservation of the incompressible aqueous solution. This parametric sweep yields a dataset of 7,332 concentration profiles.

In the second step, we use this extensive dataset to extrapolate the behaviour of interconnected heat flow chambers. For this, we set up the structure of the chamber system using a self-made LabVIEW

program that fills a 3D array (see NetworkSimulation_v3-1.llb with the settings shown in GridSimSettings.png and the program state saved in MCA31_2023_07_24.dat), representing $Z$ systems of a 2D matrix of chambers. The program assigns one inflow and a maximum of two outflow channels to each of the chambers in a random fashion. Each chamber can only connect to chambers inside the same row or one row above or below, simulating a realistic system. The inflow and outflow rates are set up according to the mass conservation of the solvent. After setting the temperature difference $\Delta T$ and defining all species $i$ of a mixture by setting $D_i$ and $S_{T,i}$, every chamber of the system has a fully defined set of boundary conditions that can be retrieved from the previously calculated dataset. Solutions of parameter values ($\Delta T$, $D_i$, $S_{T,i}$, $Q_{in}$, $Q_{out,bot}$) that are set between calculated parameters are linearly interpolated from the above dataset. The concentration profiles of each chamber $c_{m,n,i}$ at matrix position $m$ and $n$ are then renormalized column by column with the total inlet concentration $c_{in,m,n,i} = \frac{\sum_l c_{in,m,n,i,l} \times Q_{in,l}}{Q_{in}}$, with the concentrations $c_{in,m,n,i,l}$ and volume rates $Q_{in,l}$ ($Q_{in} = \sum_l Q_{in,l}$) of each of the $l_{max}$ inflow channels $l$. The outflow concentrations $c_{out,m,n,i}$ of each species are calculated by the finite element simulations and assigned to the inlet concentrations $c_{in,m,n+1}$ of the next column. The renormalized averaged concentrations $c_{bot,m,n,i}$ of the bottommost 1.5 mm volume fractions are then plotted in Fig. 4e–i. The maximum pairwise enrichments shown in Fig. 4g–i are calculated by taking the median of the ten maximum concentration ratios of two species A and B: $\mathrm{median}\left(\max_{\forall\, m,n}\left(\frac{c_{bot,m,n,i=A}}{c_{bot,n,m,i=B}}\right)\right)$. The median is used to avoid overestimating enrichment ratios by numerical noise and outliers.

Owing to the routing of the connecting channels according to the above rules as well as the random error of the measured Soret coefficients, the numerical modelling of the heat flow chamber network also has associated errors. To quantify these, we repeated the respective simulations (that is, $N_x$ by $N_y$ chambers for $Z$ systems) at least three times and calculated the mean and corresponding s.d. from the resulting enrichment values as shown in Extended Data Fig. 6. To determine the random error for the Soret coefficients shown in Extended Data Table 1, we determined the correlation of the Soret coefficients for all species present in the respective mixtures between measurement replicates (Supplementary Fig. 9). The systematic error is mainly caused by small differences in the temperature gradient between the individual replicates and equally affects the Soret coefficients of all species contained in a mixture. By fitting this correlation linearly, this systematic error can thus be determined by the deviation of the slope of the linear fit to the slope 1. Accordingly, the s.d. of the fit yields the random error (for example, because of the integral determination of the HPLC peaks, fluctuations of the column temperature during the HPLC run etc.). In the repeat runs of the network modelling for the error determination, the Soret coefficients for the species involved were, therefore, randomly chosen according to a Gaussian distribution around the average value shown in Extended Data Table 1 with a s.d. corresponding to the random error determined above. Thus, the error from the Soret coefficients could be numerically propagated and shown in the error matrices in Extended Data Fig. 6.

## Glycine-dimerization experiments

For the experiments shown in Fig. 5, we closely followed the protocol for the dimerization of glycine[11].

We prepared triplicates of 200 µl of a solution containing 10 mM and 100 mM glycine and various concentrations of TMP, ranging more than three orders of magnitude (0.10, 0.25, 0.63, 1.6, 4.0, 10, 25, 63, 158 mM). The mixture was then adjusted to pH 10.5 using NaOH and heated at 90 °C for 16 h. Using the LC protocol described above, we then analysed the yield of dimerization by comparison against standards.

The same was done for a mixture of 100 mM glycine and 100 mM TMP with measurements at different time points between 0 and 120 h.

## Modelling of reactions for network extrapolation

To model the dimerization of glycine driven by TMP in large networks of connected heat flow chambers, we first determined all required rate constants $k_1$–$k_5$ of the simplified reaction scheme[11,12]:

$$\mathrm{Gly} + \mathrm{TMP} \xrightarrow{k_1} \mathrm{GlyAct} \tag{9}$$

$$\mathrm{GlyAct} \xrightarrow{k_2} \mathrm{Gly} \tag{10}$$

$$\mathrm{GlyAct} + \mathrm{Gly} \xrightarrow{k_3} \mathrm{GlyGly} \tag{11}$$

$$\mathrm{TMP} \xrightarrow{k_4} \mathrm{waste} \tag{12}$$

$$\mathrm{GlyGly} \xrightarrow{k_5} \mathrm{Gly} + \mathrm{Gly} \tag{13}$$

Here the single glycine in equation (9) is first phosphorylated (activated) by TMP. The activated glycine can then react with another glycine in equation (11) and form the GlyGly dimer. Both the activated glycine and the glycine dimer can degrade to glycine by hydrolysis (equations (10) and (13)). For TMP, a possible hydrolysis was also taken into account (equation (12)). For the determination of $k_1$–$k_5$, we measured the product concentrations for a range of initial concentrations ($[\mathrm{Gly}]_{init} = 10$ mM, 100 mM at $[\mathrm{TMP}]_{init} = 100$ mM, $T = 90$ °C, initial pH 10.5) at different time points (Fig. 5c and Supplementary Table 17). We determined the rate $k_4$ separately with a solution of only $[\mathrm{TMP}]_{init} = 0.2$ mM in water under the same temperature and pH conditions to allow a better stability of the numerical fit described below for the remaining reaction constants $k_1$–$k_3$, $k_5$ and measured the decrease in TMP concentration over time (Supplementary Fig. 13 and Supplementary Table 19).

For the determination of reaction rates, either the full reaction system (equations (9)–(13)) or only the TMP hydrolysis (equation (12)) was implemented as a differential equation system in a 0D model (COMSOL 5.4, filename: MultiDFitGlyRctnModel_20230714_Modellv1.mph, yielding the data file 2023_07_14_MultiDFit_modelV1.dat, analysed with FreeFitter_V4-6.llb):

$$\frac{d[\mathrm{Gly}]}{dt} = -k_1 \times [\mathrm{TMP}] \times [\mathrm{Gly}] - k_3 \times [\mathrm{GlyAct}] \times [\mathrm{Gly}] \\ + k_2 \times [\mathrm{GlyAct}] + k_5 \times 2 \times [\mathrm{GlyGly}] \tag{14}$$

$$\frac{d[\mathrm{GlyAct}]}{dt} = -k_2 \times [\mathrm{GlyAct}] + k_1 \times [\mathrm{Gly}] \times [\mathrm{TMP}] \\ - k_3 \times [\mathrm{GlyAct}] \times [\mathrm{Gly}] \tag{15}$$

$$\frac{d[\mathrm{GlyGly}]}{dt} = +k_3 \times [\mathrm{GlyAct}] \times [\mathrm{Gly}] - k_5 \times [\mathrm{GlyGly}] \tag{16}$$

$$\frac{d[\mathrm{TMP}]}{dt} = -k_1 \times [\mathrm{Gly}] \times [\mathrm{TMP}] - k_4 \times [\mathrm{TMP}] \tag{17}$$

The solution of this reaction system was then solved for the initial concentrations $[\mathrm{Gly}]_{init} = 100$ mM and 10 mM and $[\mathrm{TMP}]_{init} = 0.0001$ M, 0.000251189 M, 0.000630957 M, 0.00158489 M, 0.00398107 M, 0.01 M, 0.0251189 M, 0.0630957 M and 0.158489 M (corresponding to the experiment in Fig. 5c) under variation of the rate constants (Supplementary Table 18). The solution concentrations were determined for the reaction times 0–122 h. For the separate determination of $k_4$ mentioned above, the initial concentration $[\mathrm{TMP}]_{init} = 0.2$ mM was chosen according to the experiment (Supplementary Fig. 13). With

the extensive datasets obtained in this way, we were able to simultaneously fit all of the experimental data points to a common parameter set with a custom-made LabVIEW program using a Levenberg–Marquardt algorithm (see FreeFitter_V4-6.llb). The rates obtained in this way are: $k_4 = 3.5 \times 10^{-7}$ s$^{-1}$ ($\pm 19\%$) and $k_1 = 1.0 \times 10^{-3}$ Ms$^{-1}$ ($\pm 41\%$), $k_2 = 1.1 \times 10^{-4}$ s$^{-1}$ ($\pm 10\%$), $k_3 = 9.7 \times 10^{-5}$ Ms$^{-1}$ ($\pm 21\%$) and $k_5 = 1.2 \times 10^{-6}$ s$^{-1}$ ($\pm 20\%$). Although the error bars in Fig. 5c indicate the s.d. from triple replicate experiments, we have determined the error of the modelling described above by a stochastic approach. We calculated the resulting product concentrations 500 times for the fitted set of rate parameters, choosing the rates from the fit using a weighted random number generator with a Gaussian probability distribution with the s.d. according to the previously given rate error. The shaded area contains 68.27% of all these 500 runs, equivalent to one s.d. To determine the reaction yield in the network of interconnected heat flow chambers, we proceeded as described next.

Using the rate constants thus obtained, we calculated another dataset using COMSOL as the result of the equations (14)–(17), in which we varied the initial concentrations [Gly]$_{init}$ and [TMP]$_{init}$ over a range of $1 \times 10^{-10}$ M to 1 M each, with five concentrations per decade (see GridReactionGlyRctnModel_20230717_Modellv1.mph, resulting in data file 2023_07_17_ModelV1ForGridSimFig5.dat for analysis with FreeFitter_V4-6.llb). The product concentrations were determined over a reaction time of 120 h (divided into 20 time points). We were thus able to map the TMP and Gly concentrations obtained from network modelling described above (using the Soret coefficients from Extended Data Table 1) in each individual heat flux chamber to this reaction dataset and thus determine the amount of product obtained and the reaction yield (Fig. 5d–f). The errors given in Fig. 5e,f were calculated stochastically as described above by recalculating the network model ten times, choosing the Soret coefficients with a weighted random generator with Gaussian probability distribution at one s.d. of the previously determined random error.

The network continuously provides the reactants according to the accumulation characteristics shown in Fig. 4, taking into account the high Soret coefficient of TMP $S_{T,TMP} \approx 7 \times 10^{-3}$ K$^{-1}$. As shown in Fig. 5c, right, the maximum reaction yield is already reached after 16 h, but the relaxation time of a chamber with throughflow in the range $Q_{flow} = 1$–10 nl s$^{-1}$ is expected to be on the order of Volume$_{chamber}/Q_{flow} \approx 10^{1}$–$10^{2}$ h. Therefore, to sufficiently account for the hydrolysis included in the model, we calculate the product yield after 120 h reaction time. The reactions are each assumed to occur in a reaction volume at the bottom of the respective chamber in which the reactants are most concentrated (Supplementary Fig. 11a).

## Data availability

All data generated or analysed during this study are included in this published article (and its Supplementary Information Files). A comprehensive, ready-to-use dataset to the supplied code is included in the Supplementary Code and Data in the form of dat files as described in Methods and can be loaded directly from the LabVIEW programs. Source data are provided with this paper.

## Code availability

The full details of the finite element simulation from Figs. 4 and 5 and its analysis tools are supplied as model and LabVIEW files in the Supplementary Code and Data and described in Methods.

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

**Acknowledgements** We thank K. Le Vay for proofreading the manuscript, J. Langlais, M. Rappold, A. Schmid and M. Weingart for experimental support and N. Yeh Martín, S. Rout, A. Kühnlein and I. Smokers for fruitful discussions. This work was financed by the Volkswagen Foundation initiative 'Life? – A Fresh Scientific Approach to the Basic Principles of Life' (T.M., D.B. and C.B.M.) and by the Deutsche Forschungsgemeinschaft (DFG, German Research Foundation) under project ID 364653263 – TRR 235 (T.M., P.A., B.S., D.B. and C.B.M.) and under Germany's Excellence Strategy – EXC-2094 – 390783311 (T.M., D.B. and C.B.M.). Funding from the Simons Foundation (327125 to D.B.) and from the European Research Council EvoTrap #787356, ERC-2017-ADG (P.A. and D.B.) is gratefully acknowledged. This work was supported by the Center for NanoScience (CeNS) in Munich.

**Author contributions** T.M., P.A. and C.B.M. conceived and designed the experiments. T.M., P.A. and C.B.M. performed the experiments. T.M., P.A. and C.B.M. analysed the data. T.M., P.A., B.S., D.B. and C.B.M. wrote the paper. All authors discussed the results and commented on the manuscript.

**Competing interests** The authors declare no competing interests.

**Additional information**
**Correspondence and requests for materials** should be addressed to Christof B. Mast.

**a**: Heat flux cell assembly **b**: ΔT implementation **c**: Differential analysis

**Extended Data Fig. 1 | Experiment setup and analysis. a**, Preparation of heat flux cells. The microfluidic structure defined by the FEP foil (**5**) is sandwiched between two sapphires with thicknesses of 500 μm (**4**; cooled sapphire, with inlets/outlets of 1 mm diameter) and 2,000 μm (**6**; heated sapphire). The sapphire–FEP–sapphire block is then placed on an aluminium base (**2**) covered by a heat-conducting foil on the back (**1**) and front (**3**) for optimal heat conduction to the cryostat and the sapphire, and held in place by a steel frame (**7**). The steel frame is connected to the aluminium base by six torque-controlled screws for a homogeneous force distribution. The height of the chamber is measured with a confocal micrometer at three positions (bottom, middle and top) to ensure a homogeneous thickness. Together with another heat-conducting foil (**8**), an Ohmic heating element (**9**) is placed on top of the heated sapphire mounted to the steel frame with torque-controlled screws. Chambers are pre-flushed using low-viscosity, fluorinated oil to check for tightness and push out residual gas inclusions. The sample is then pulled into the oil-filled chamber. After loading the sample, the tubings are closed. **b**, Application of the temperature gradient. The assembled chamber is mounted onto a cooled aluminium block connected to a cryostat. The heaters are connected to a power supply that is controlled by Arduino boards. To stop the experiment, heaters and the cryostat are turned off and the chamber is stored at −80 °C for at least 15 min. **c**, Differential recovery of four fractions. The freezing allows us to cut the frozen interior of the heat flow cell into four fractions. Fewer fractions would lower resolution by averaging over a larger fraction of the chamber, whereas more fractions would make subsequent analysis difficult because of volume limitations and being more prone to error (Extended Data Fig. 2).

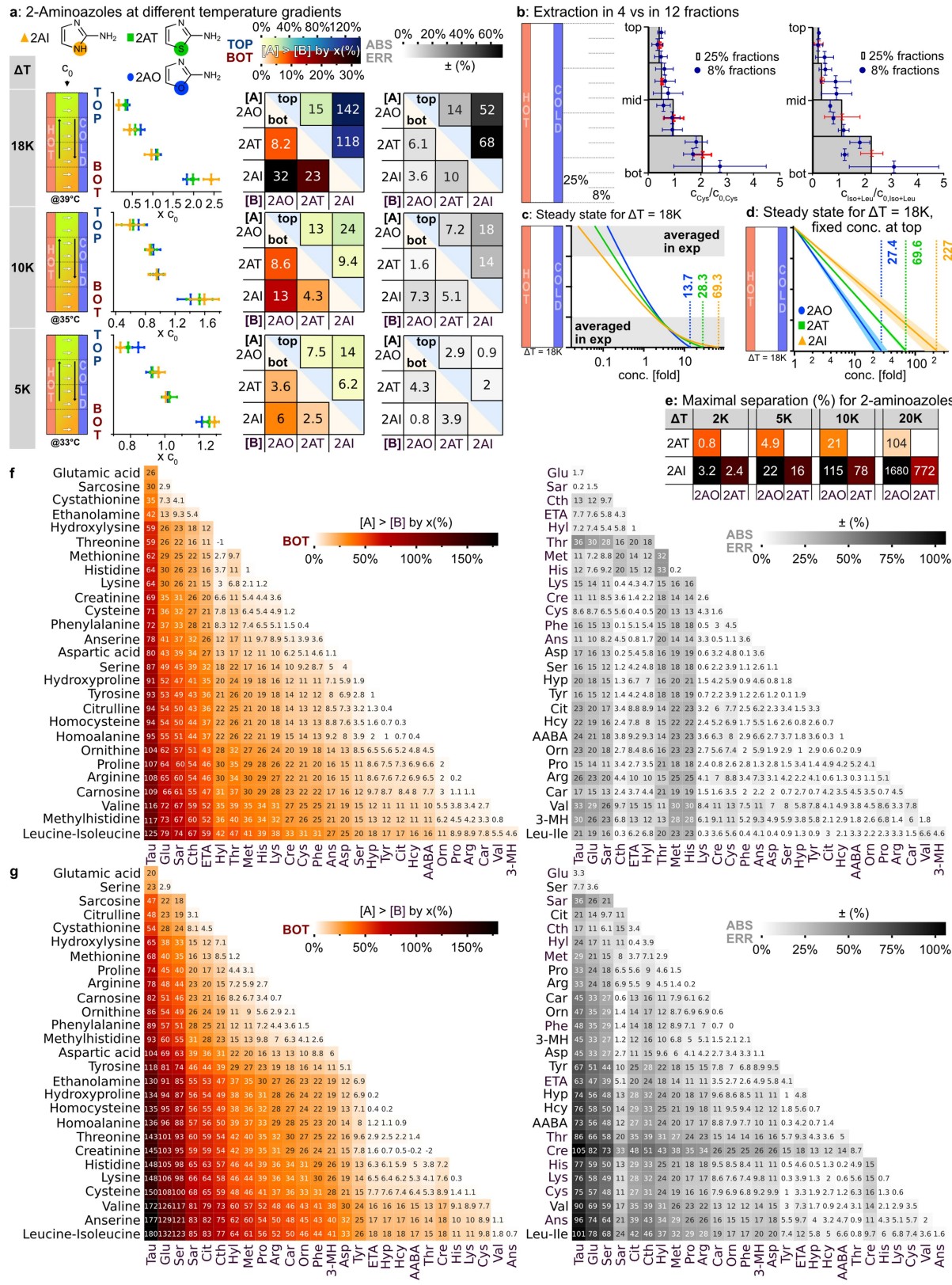

**Extended Data Fig. 2** | See next page for caption.

**Extended Data Fig. 2 | Enrichment of 2-aminoazoles and amino acids for different temperature gradients (errors = s.d., three repeats, same for all; see raw data tables in Supplementary Tables 5–65). a**, Accumulation plots for 2-aminoazoles under 5 K, 10 K and 18 K gradients. For 18 K, accumulation at the bottom is strongest (up to 2.5-fold $c_0$). The strong accumulation for high thermal gradients goes together with an increased separation between species, both at the bottom, for which 2AI is most abundant, and at the top, for which the situation is reversed with up to 142% excess of 2AO versus 2AI. Corresponding errors are shown in the right column. The error maps each show the absolute enrichment error, for example, 2AI versus 2AO (32±3.6)%, so the error range varies between (32−3.6)% = 28.4% and (32+3.6)% = 35.6%. **b**, Extraction of cysteine and isoleucine + leucine in four (25%) and 12 (8%) fractions. The 8% bottom fraction shows, as expected from theory, a higher value but has a higher error. Also, spatial errors ($y$ axis) increase on more detailed extraction. **c**, Calculated steady state for 2-aminoazoles for d$T$ = 18 K in a closed heat flow chamber. The experimentally average zones (top 25% and bottom 25%) are marked in grey. In the bottom fraction, the average goes over zones of different concentration ratio, including both excess and underrepresentation of species and, thus, lowering the effectively measured enrichment. This indicates that enrichments in a realistic setting can go up further. **d**, Assuming a fixed concentration at the top, we find that concentrations vary more than one order of magnitude. **e**, We use this as a measure of maximal pairwise separation between species for several temperature gradients and species (Supplementary Fig. 6), finding that, even at 5 K, separation values >20% are possible. **f,** In a mixture of non-proteogenic and proteogenic amino acids and small organics molecules (diluted to 50 µM and containing 0.02-fold lithium citrate, 0.01% phenol and 0.2% thiodiglycol), no clear pattern of separation is observed with non-proteogenic amino acids on both the strongly and the weakly accumulating sides. Error maps defined as in **a. g**, The same is the case for extraction in 8% fractions, with higher absolute values but also increased errors owing to the now higher impact of volume variations when manually cutting the frozen chamber content. Error maps defined as in **a.**

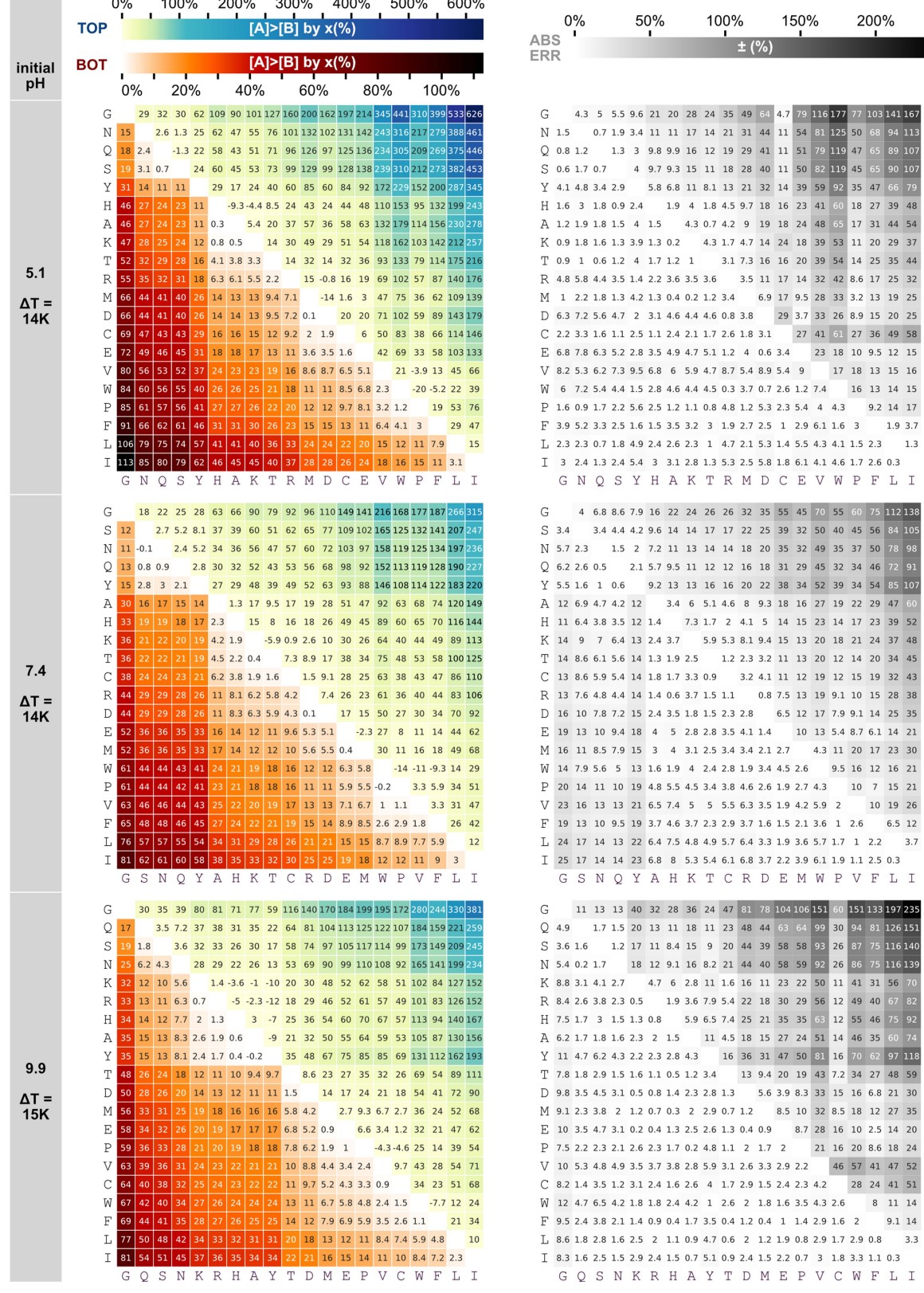

**Extended Data Fig. 3 | Enrichment between proteinogenic amino acids (30 µM each) with different initial pH values (errors = s.d., three repeats, same for all).** Analysis reveals a strong enrichment of amino acids isoleucine, leucine and phenylalanine at the bottom of the chamber for all pH values against glycine (65–113%) and serine, asparagine and glutamine (35–85%).

In the top fraction, the situation is inverted, with up to 300+%. The order and enrichments change with pH as thermophoresis is influenced by, for example, charged state. On the right side, the errors per value of the heat maps are shown as defined in Extended Data Fig. 2a.

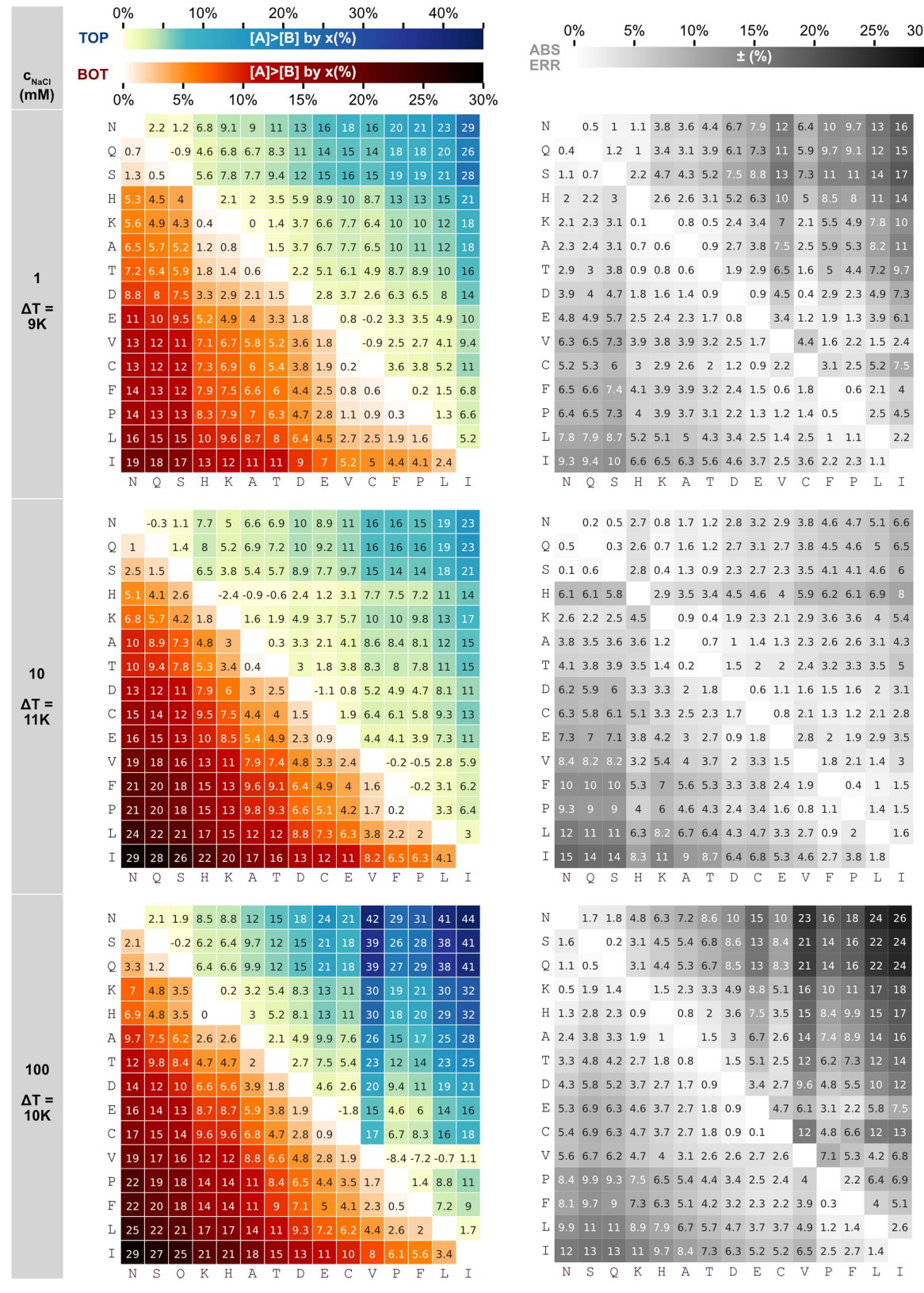

**Extended Data Fig. 4 | Enrichment between proteinogenic amino acids (30 μM each) with different concentrations of NaCl (errors = s.d., three repeats, same for all).** For all salt concentrations, analysis reveals a strong enrichment of amino acids isoleucine and leucine at the bottom of the chamber against asparagine, serine and glutamine (13–29%). In the top fraction, the situation is inverted, with up to 40+%. On the right side, the errors per value of the heat maps are shown as defined in Extended Data Fig. 2a.

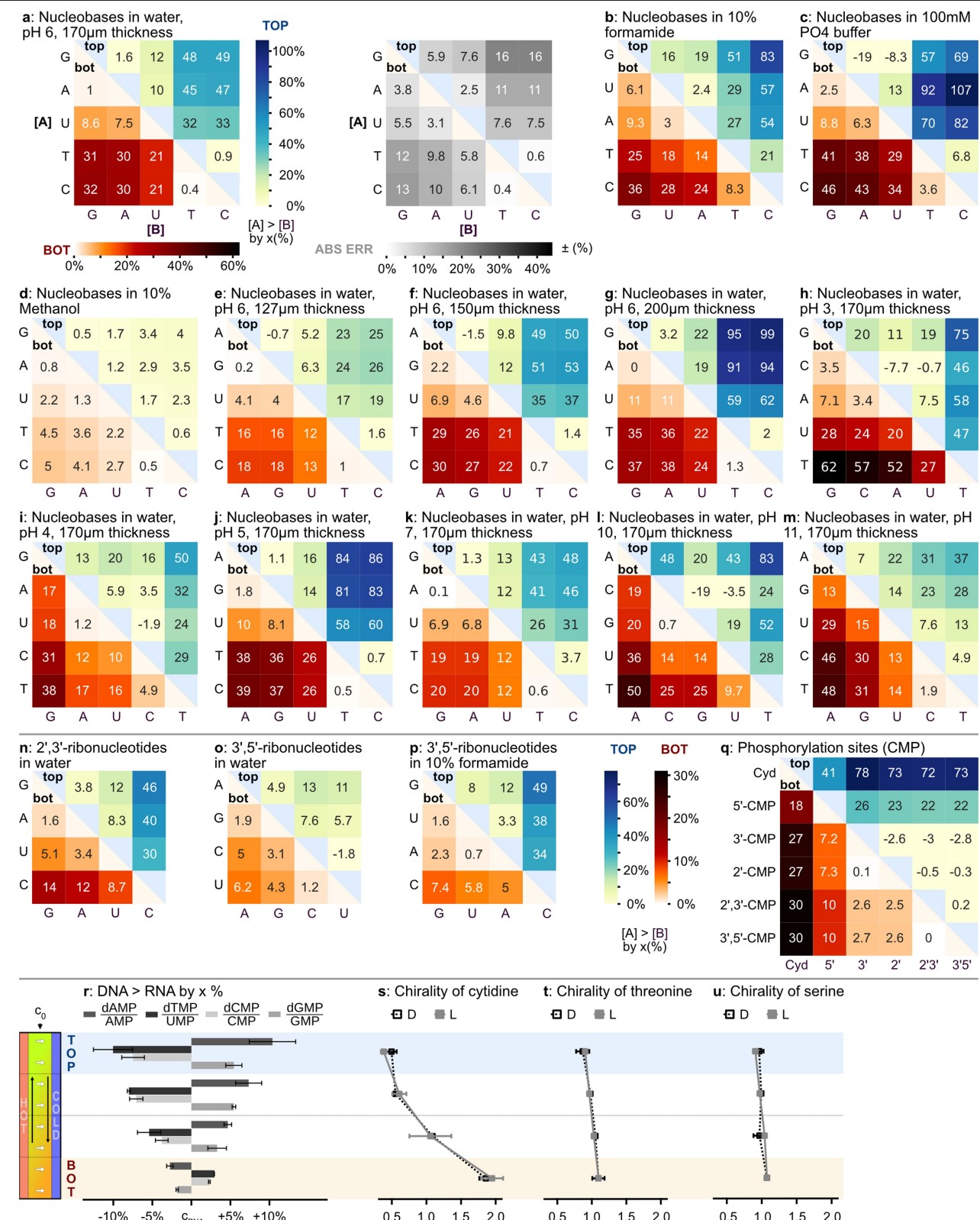

**Extended Data Fig. 5 |** See next page for caption.

**Extended Data Fig. 5 | Enrichment of nucleobases and nucleotides for various settings (errors = s.d., three repeats, same for all). a**, Plot from Fig. 3 as reference, with its corresponding error map as defined in Extended Data Fig. 2a. **b**–**d**, Enrichments in different solvents. Although 10% formamide and phosphate buffer do not alter accumulation and separation, for methanol, the amplitude decreases massively. **e**–**g**, Enrichment for different fracture thicknesses. After 18 h, accumulation is stronger for thicker cells, even though this also goes together with varied times needed to reach a steady state[46]. **h**–**m**, Enrichments for different initial pH values. The general behaviour and order of magnitude stay the same as shown in Fig. 3c. However, the most dominant species changes over pH. Error maps for **b**–**m** are shown in Supplementary Fig. 3. **n**–**p**, Enrichment of 2′,3′-cyclic and 3′,5′-cyclic nucleotides in water and for the latter in 10% formamide in water. For 2′,3′-cyclic nucleotides, C is enriched up to 14% over the other nucleotides. For 3′,5′-cyclic nucleotides, we find that C (bottom) and G (top) are dominant in individual fractions in formamide, whereas in water, overall enrichment is weaker and dominated by A and U. **q**, Enrichment between cytidine and cytidine monophosphates (CMP, 30 μM each). Nucleotides (5′, 3′, 2′, 2′,3′, 3′,5′) are enriched against the nucleoside by at least 18%. Cyclic CMPs are enriched against linear CMPs by up to 10%. Even though having the same mass, 2′-CMP is enriched against 5′-CMP and 3′-CMP by 2.5%. Errors for **n**–**q** are shown in Supplementary Fig. 3. **r**, Direct comparison of DNA versus 5′-RNA nucleotides. For pyrimidine nucleotides, we observe a stronger accumulation at the bottom for DNA nucleotides (dCMP versus 5′-CMP and dTMP versus 5′-UMP). For purine nucleotides, the inverse situation is true with up to 10% DNA nucleotide excess at the top (dAMP versus 5′-AMP). **s**–**u**, Accumulation profiles of D-cytidine and L-cytidine, D-threonine and L-threonine and D-serine and L-serine. We do not find any substantial enrichment between the two chiralities.

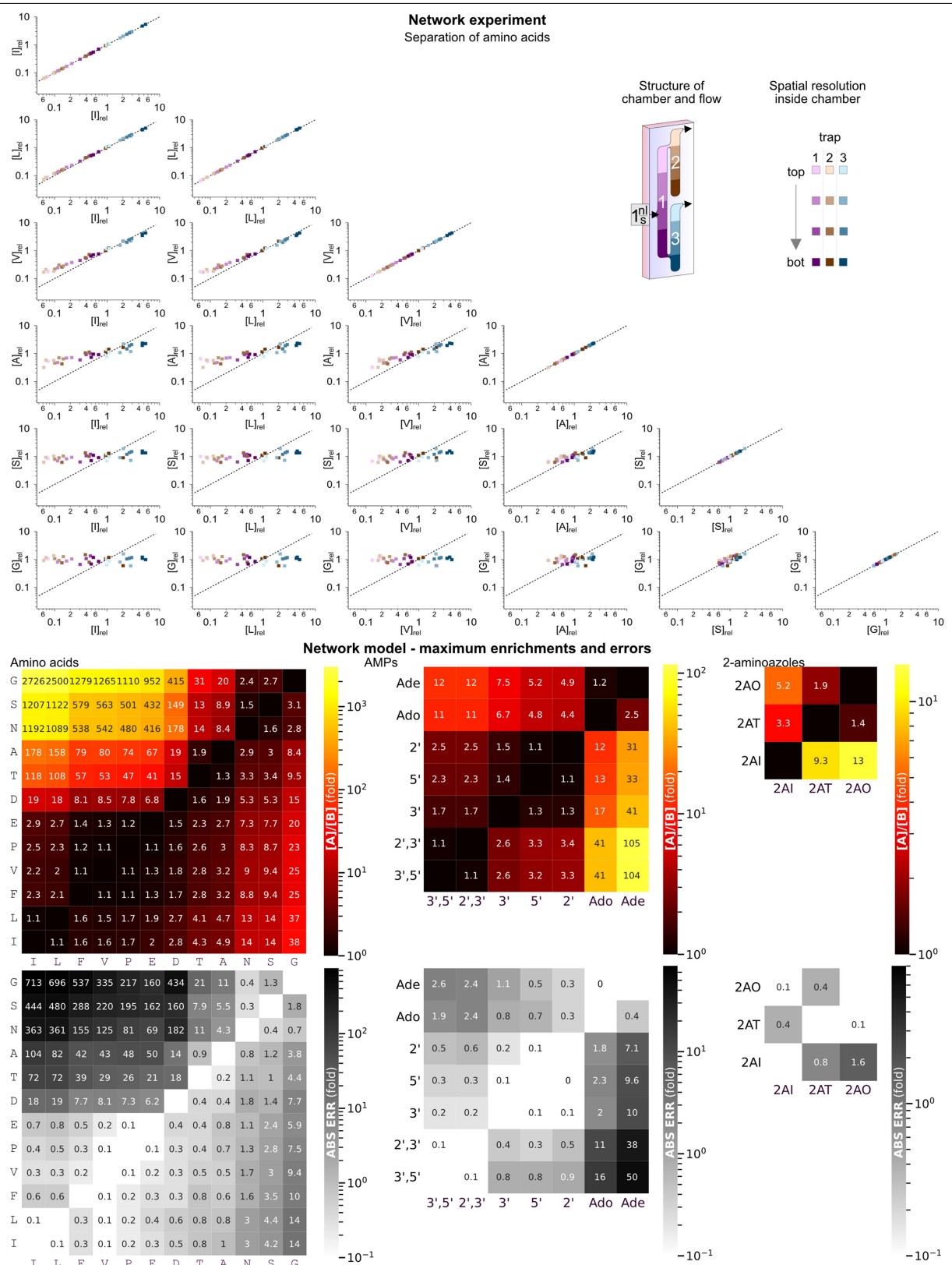

**Extended Data Fig. 6 | Experimental and modelled separation of molecules in a network of connected rock cracks. Top**, experimental setup of a small network of three interconnected chambers with a volume inflow of 1 nl s⁻¹ of the same amino acid mixture used in Fig. 2 and $\Delta T = 16$ K. After 60 h, the chamber contents from three repeats were frozen and divided into individual parts according to the colour gradations and measured by HPLC. As well as the main-text examples (I versus N and I versus F), selected pairs are shown.

For instance, thermophoretically different amino acids glycine (G) and isoleucine (I) separate readily, whereas mass-identical L and I only show minor concentration differences in our experimental system. Further examples and a detailed overview over spatial resolution are shown in Supplementary Figs. 4 and 5. **Bottom**, maximum enrichments in the system as shown in Fig. 4d for mixtures of amino acids, adenine (nucleosides/nucleotides), and 2-aminoazoles. Error maps as defined in Extended Data Fig. 2a.

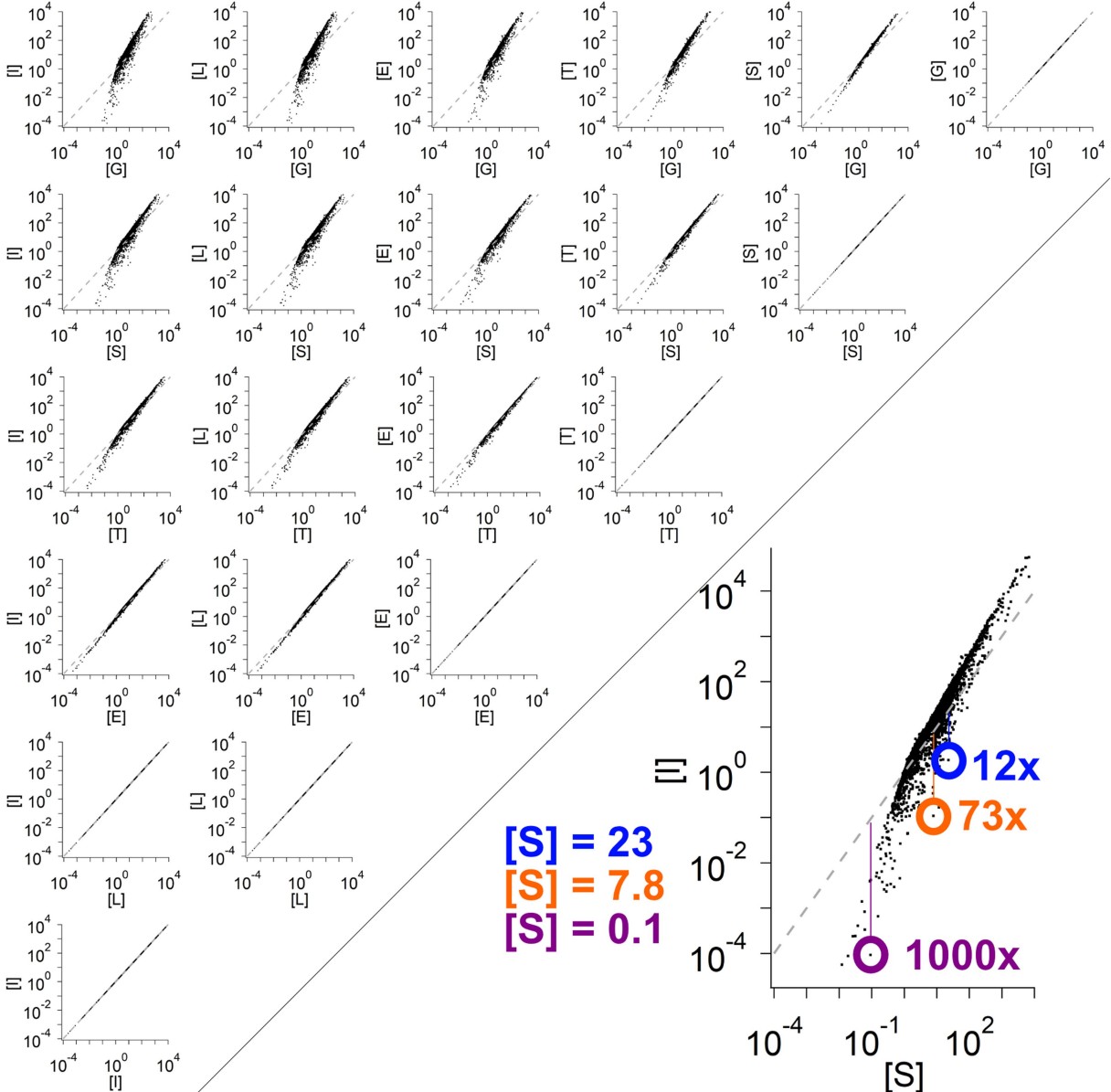

**Extended Data Fig. 7 | Pairwise enrichment of amino acids from extracts of Fig. 4g, analogous to Fig. 4e.** Each scatter plot shows the correlation of concentrations in the lower part of each heat flow chamber of 30 simulated networks of 20 × 20 chambers, normalized to the inflow concentration (= 1). Amino acids with similar thermophoresis (for example, G versus S) are similarly concentrated (or depleted) and, thus, hardly enriched against each other. Amino acids with strongly different thermophoresis (see Extended Data Table 1), such as I versus S, are concentrated or depleted very differently. The thermophoretically stronger species I is, thereby, more strongly depleting in the upper regions of the network than the thermophoretically weaker S, so that S occurs here, for example, with more than 1,000 times the concentration of I. The absolute concentration of S is still 0.1 times the initial concentration (purple). Even with an enrichment of 73 times S compared with I (orange), the absolute concentration of S is still at a usable 7.8 times the initial concentration.

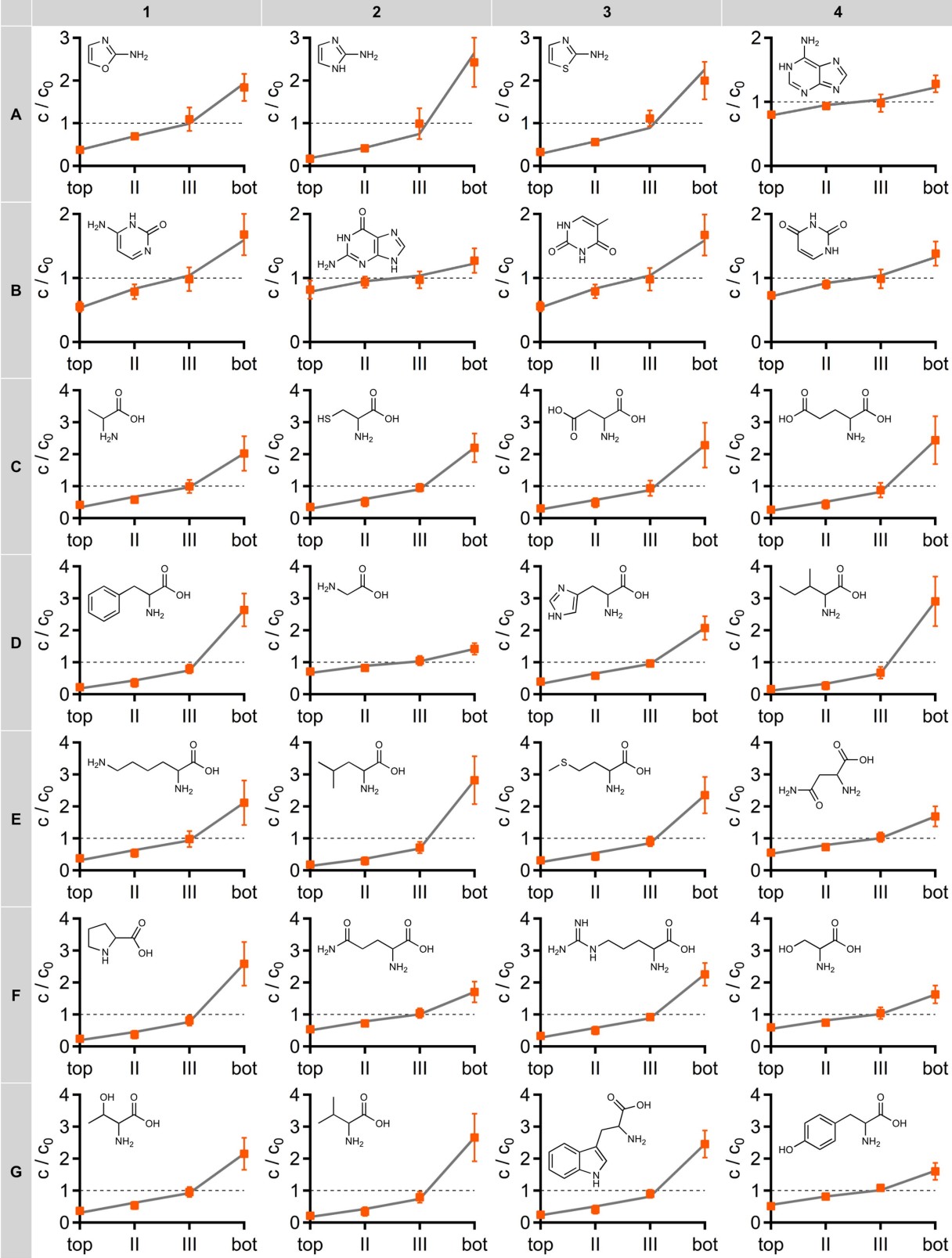

**Extended Data Fig. 8 | Concentration profiles in heat flux cell and fit for Soret coefficients.** For aminoazoles (A1–A3), nucleobases (A4–B4) and amino acids (C1–G4), the measured concentration/initial concentration is depicted for all four fractions (errors = s.d., three repeats, same for all). Further data are shown in Supplementary Fig. 7. For fitted Soret coefficients, see Extended Data

Table 1. **A1**: 2AO; **A2**: 2AI; **A3**: 2AT; **A4**: adenine; **B1**: cytosine; **B2**: guanine; **B3**: thymine; **B4**: uracil; **C1**: alanine; **C2**: cysteine; **C3**: aspartic acid; **C4**: glutamic acid; **D1**: phenylalanine; **D2**: glycine; **D3**: histidine; **D4**: isoleucine; **E1**: lysine; **E2**: leucine; **E3**: methionine; **E4**: asparagine; **F1**: proline; **F2**: glutamine; **F3**: arginine; **F4**: serine; **G1**: threonine; **G2**: valine; **G3**: tryptophan; **G4**: tyrosine.

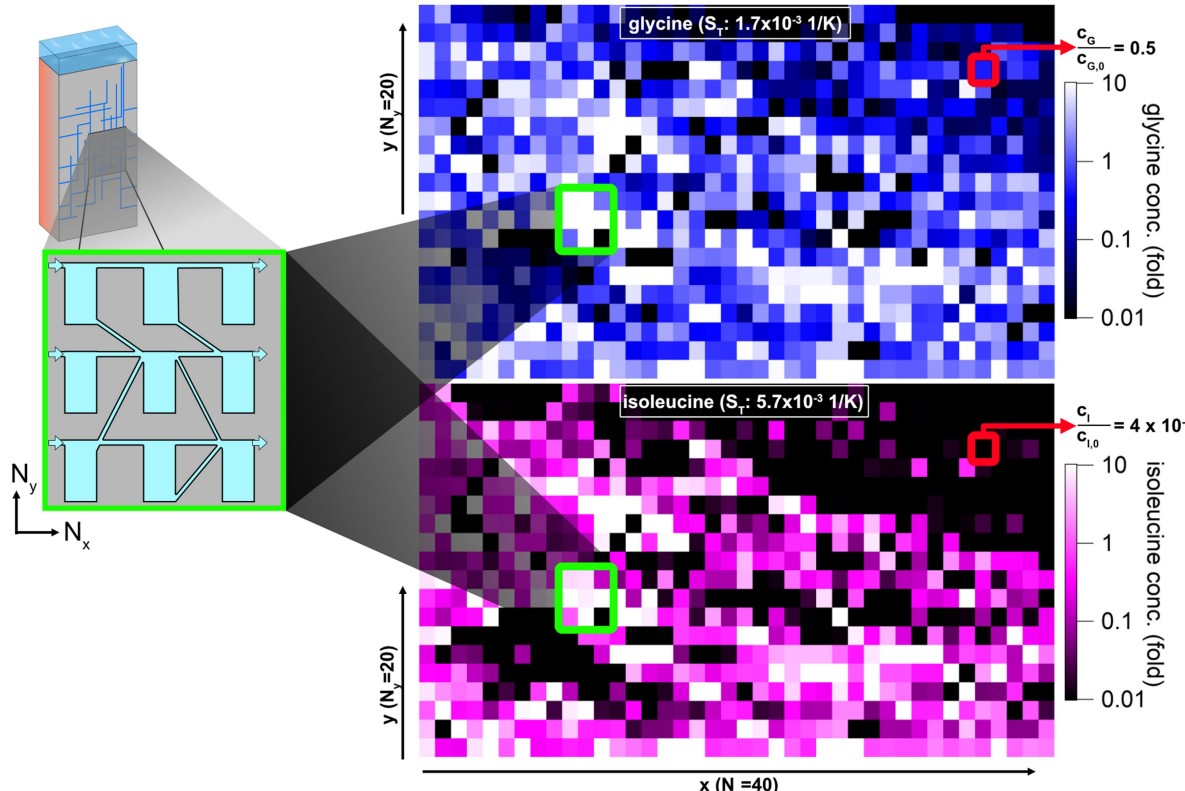

**Extended Data Fig. 9 | Numerical calculation of an exemplary accumulation of substances with low thermophoretic strength (for example, glycine) versus those with high thermophoretic strength (for example, isoleucine).** The 2D plots show, respectively, glycine (blue) and isoleucine (purple) bottom concentration for the same system of $N_x = 40$ by $N_y = 20$ randomly connected heat flow chambers (in a single system). Each pixel corresponds to one chamber. Isoleucine is more concentrated because of its high Soret coefficient, so there are many chambers with more than tenfold concentration. Owing to the resulting more efficient transport to the lower part of the system, it is only present in low concentrations further downstream in the upper part of the system (red box, 0.0004-fold initial concentration). Glycine is less concentrated owing to the lower Soret coefficient and, thus, less transported to the lower part of the system. Downstream, it is therefore still found at higher concentrations in the upper part of the system (red box, 0.5-fold initial concentration). In summary, substances with high thermophoretic strength are purified in the front part ($x < 20$, upstream) of the system and substances with low thermophoretic strength in the back part ($x \geq 20$, downstream).

**Extended Data Table 1 | Experimentally determined Soret coefficients for all species**

| | $S_T$ ($10^{-3}$ 1/K) | +- % | | $S_T$ ($10^{-3}$ 1/K) | +- % |
|---|---|---|---|---|---|
| 2AO (51 °C) | 3.4 | 8 | C (40 °C) | 3.1 | 36 |
| 2AT | 3.9 | 12 | A | 1.5 | 27 |
| 2AI | 5.5 | 13 | G | 1.4 | 36 |
| | | | U | 1.9 | 30 |
| | | | T | 3.1 | 36 |
| H (34 °C) | 3.2 | 21 | Cyd (40 °C) | 2.3 | 51 |
| N | 2.2 | 17 | Urd | 1.9 | 47 |
| S | 2.2 | 12 | Ado | 1.7 | 46 |
| Q | 2.3 | 17 | Guo | 1.7 | 44 |
| R | 3.8 | 21 | 5'-CMP (40 °C) | 5.3 | 23 |
| G | 1.7 | 9 | 5'-AMP | 4.8 | 23 |
| D | 3.8 | 23 | 5'-UMP | 5.0 | 23 |
| E | 4.2 | 26 | 5'-GMP | 5.1 | 24 |
| T | 3.4 | 23 | 2',3'-CMP (40 °C) | 6.5 | 13 |
| A | 3.1 | 21 | 2',3'-UMP | 5.6 | 10 |
| P | 4.5 | 26 | 2',3'-AMP | 5.3 | 17 |
| C | 3.5 | 23 | 2',3'-GMP | 5.2 | 10 |
| K | 3.4 | 22 | 3',5'-CMP (40 °C) | 5.5 | 21 |
| M | 4.1 | 23 | 3',5'-AMP | 5.1 | 20 |
| V | 4.7 | 30 | 3',5'-GMP | 5.3 | 18 |
| I | 5.7 | 30 | 3',5'-UMP | 5.6 | 18 |
| L | 5.4 | 29 | 2'-AMP (40 °C) | 3.7 | 21 |
| F | 4.8 | 26 | 3'-AMP | 3.4 | 21 |
| Y | 2.1 | 25 | 2'-CMP (40 °C) | 4.7 | 14 |
| W | 4.5 | 19 | 3'-CMP | 4.6 | 14 |
| | | | dCMP (31 °C) | 5.0 | 12 |
| | | | dAMP | 4.5 | 14 |
| TMP | 7.5 | 43 | dGMP | 4.6 | 12 |
| | | | dTMP | 5.0 | 12 |

Temperature of the mass centre is given for each group of molecules (errors=s.d., three repeats, same for all). For the fitting routine and simulation, see Methods. The fitted experimental data are shown in Extended Data Fig. 8 and Supplementary Fig. 7.