## [Peer Review File · Nature]

Manuscript Title: Heat flows enrich prebiotic building blocks and enhance their reactivity

Reviewer Comments & Author Rebuttals

Reviewer Reports on the Initial Version:

Referees' comments:

Referee #1 (Remarks to the Author):

Mast and co-workers report on the separation of mixtures of prebiotically relevant building blocks through heat flows in confined volumes. This is very exciting work. Although I have several questions (see below), the work is impressive and potentially of very high relevance for the field of prebiotic chemistry, where no convincing, general methods exist that could yield highly enriched mixtures of starting materials for chemical reactions. This work demonstrates that such a method might indeed exist in the ubiquitous cracks in rocks.

My first main point of criticism is that I struggle with the presentation of the data. For example: page 4: In the top fraction, 2AI is depleted by 40 % compared to \bar{c}_{top} while 2-aminooxazole (2AO) is enriched by 40 %. Upon direct comparison, this corresponds to an enrichment of 2AO over 2AI by 142 % = $\bar{c}_{2AO,top}/\bar{c}_{2AI,top} - 1$ in the top fraction (Extended Data Fig. 2).

- I might be missing something, but because the authors did not provide the actual concentrations measured, it is very difficult to check this calculation. By also not providing the numbers for the average concentrations in the top and bottom fractions, it is difficult to appreciate the magnitude of the effect. I am not suggesting the data is wrong, but the current presentation makes it (for me) impossible to follow.

- Why is the label in Figure 2a $c_0/2$ in the top, but $2c_0$ in the bottom fraction?

A second main concern is the reliability of the data. In Extended data Fig 2, the absolute error associated with the 142% is 52%. What does this mean? I would expect the absolute error to be the difference between the measured value and the actual value. But here it is normalized against the average concentration.

- In the same context, on page 5 the authors state: Thus, a single heat flow chamber can already build up significant relative concentration differences." . What do the authors mean with significant. It seems to me that the large error would indicate that statistical significance is often not reached?

The concern about reliability/accuracy is especially important because the presumed prebiotic effect on the separation of species in rocks will be much larger than the separation effect seen in a single chamber. But all experiments are on single chambers and these values are then used as input for the simulations. The authors should provide realistic confidence intervals for the simulations.

- Accordingly, for 2'- versus 3'- and 5'-AMP, a maximal enrichment of 35 % is achieved in the lowest micrometer instead of the 4 % obtained by averaging over a quarter of the volume (Fig. 2d) This is very cryptic. Did the authors MEASURE this, or is it a result of a calculation (I appreciate that the bottom micrometer is most concentrated, but how do the authors get to these numbers?)
- The data in Figure 3 (ignoring the difficulty I have discussed above with respect to interpreting the actual numbers) are very interesting, as they show that AAs separate strongly in these thermophoretic chambers. Have the authors attempted to separate all AAs in one single experiment? It would be very interesting to see if a similar degree of separation would be achieved, or that interactions between the AAs would diminish or enhance the separation. This would be a very important experiment to do! The prebiotic relevance of the work is to a large extent built on the need for purification of reagents prior to a reaction. Therefore, the reagents must be enriched from very complex mixtures (not pairwise...).
- The simulations in Figure 4 are very elegant and provide a realistic insight into how mixtures might be separated in rocks with multiple cracks, each with a temperature gradient. Can the authors discuss how relevant a value of 2K for the temperature gradient is? Would this be a lower estimate? I can well imagine that the temperature difference in a microporous rock is 1K or even just a fraction of a degree across a single crack?
- Also: for the simulations, did the authors really use a flow rate in the range of 10^{-3} nL per s??

My second main point of criticism concerns the (predicted) increase in conversions of chemical reactions in networks of cracks. I think these modeling results are not sound.

- First, for the reaction between cytidine and TMP, the model cannot use an equilibrium equation. This reaction is irreversible. Second, the dimerization of glycine is not a trimolecular reaction. Instead, first one glycine is activated via phosphorylation with TMP and then the second glycine reacts with the activated intermediate. Thirdly, I don't think one can/should use the product-dependent reaction rate, as there is no 'product inhibition'.
- Furthermore, considering the long reaction times, I wonder whether (or rather, suggest that) hydrolysis reactions should be taken into account. Both TMP and the phosphorylated and dipeptide products can hydrolyze under basic conditions. It might well be that any increase in yield would be eliminated by hydrolysis.

As it stands, Figure 5 should be backed up by experimental data. Now, the reactions are carried out in bulk, at least an experiment in a single trap should be done to see if there is any acceleration at the top or bottom of the trap.

Referee #2 (Remarks to the Author):

The manuscript by Matreux and colleagues discusses how prebiotically relevant molecules could be separated and enriched from complex mixtures using temperature gradients that might occur in rock

fractures. The authors used a microfluidic heat flow chamber to study the accumulation of molecules from different mixtures within this simulated rock fracture. HPLC analysis by UV and fluorescence detection was used to determine concentrations within different sections of the flow chamber. From the thermophoretic properties the accumulation and separation of complex mixtures in a network of interconnected “rock fractures” was modelled. The authors show how the accumulation of molecules might be beneficial for prebiotic reactions to boost the yields.

The authors identified an extremely important problem in prebiotic chemistry, as most reactions provide product mixtures. Side products are often not compatible with downstream reaction steps and therefore intermediates need to be purified or enriched. The work in this manuscript provides an innovative concept to potentially solve this problem.

Having said this, the work presents a good starting point but I am not yet convinced that the authors fully solved the problem. The theoretical modelling calculates a network of interconnected rock fractures, which also assumes a flow through the rock cracks. However, the authors use a fully closed heat flow chamber for experimental validation. In previous work the authors have already shown that concentration gradients can be achieved in such a system. They now transfer this concept to a much larger set of molecules and show that there are differences in the concentration gradients between the molecules. This leads to relative enrichment/depletion of compounds within a defined section of the flow cell. However, interconnected rock fractures also means that compounds need to be transferred from one rock fracture to the next, which has not been demonstrated here. Such a transfer might interfere with the convection inside the heat flow chamber, which is responsible for the concentration gradient of the tested molecules. Therefore, to validate the theoretical concept, the authors need to show how they can connect flow chambers together in order to transfer compounds by flow. This means the system is actually open with input and output as shown in Figure 1b. Here it would be important to show that connecting flow chambers can potentiate the enrichment without losing too much material. Experimental demonstration of compound separation by interconnecting several flow chambers would be enough to merit publication in nature. The demonstration of the “prebiotic chemistry” is not really needed and feels more like an add on.

I might have fully misunderstood some of the figures but it seems the provided data in this manuscript is not always consistent or missing. The data in Figure 2c and 2d does not match. In 2c (top) adenosine is actually depleted in comparison to the average concentration. In contrast, the investigated phosphorylated adenosine species (AMPs) are all enriched relative to the average concentration. Figure 2d (top), however, shows the opposite (adenosine being enriched compared to the different AMPs). A similar inconsistency is found in extended data figure 4f (bot), which states that all phosphorylated species (CMPs) are enriched in comparison with cytidine by at least 18%. A look at figure 2d, shows that only 2',3'-CMP is more strongly enriched compared to cytidine. Some of these enrichment blots suddenly include 2'-phosphorylated nucleotides but the data is nowhere to be found. In addition, the main text states “The most significant enrichment of guanine species in the bottom fraction is achieved in mixtures of nucleosides or 2',3'-phosphorylated nucleotides”. Looking at figure 2c shows that only guanosine-5'-monophosphate is slightly enriched. All other guanine species, including the nucleoside,

are depleted.

Given these inconsistencies figure 3 cannot be properly evaluated because the source data is missing. All source data (integrals from HPLC analysis, average concentrations, enrichment in % etc. should be provided for example in a table in the extended data). A representative example chromatogram of the experiments should be provided as well to check peak separation or relative peak areas.

Other comments:

- The authors usually compare the same class of molecules for enrichment. In extended data figure 4 they tested chirality as well. It would be interesting to also test sugar/nucleoside isomers (e.g. ribose vs. other pentoses or alpha vs. beta nucleosides). Separation of such compounds could have a huge impact.
- The authors state that "...the enrichment measured in the HPLC is significantly lower than present in the heat flow chamber." They assume this because they take the lower 25% fraction. To validate this claim, the authors could also use the lower 10% fraction. The lower 10% should have the largest concentration and therefore should be suitable for HPLC analysis. The concentration can then be calculated from a calibration curve and compared to the 25% sample.
- "Accordingly, for 2'- versus 3'- and 5'-AMP, a maximal enrichment of 35% is achieved in the lowest micrometer instead of the 4% obtained by averaging over a quarter of the volume (Fig. 2d)." It is not clear where the 35% are coming from but this value does not appear in Figure 2d.
- Methods: Treatment of data. The c in the equations is not defined but it seems this is the concentration? How the authors convert the HPLC peak integrals into concentration is not clear.

Referee #3 (Remarks to the Author):

Heat flows purify and boost reactivity of >50 building blocks of life

Dear Editor, dear authors,

I really enjoyed reading your paper that represents a good approximation to the study of complex processes in prebiotic chemistry studies with a novel approach.

Overview and general recommendation

In prebiotic chemistry studies, the synthesis of many basic molecules for living beings has been achieved with great success. However, the synthesis is only the first step in the complex process that led to the emergence of the first organisms. When any chemical synthesis is carried out, an enormous variety of products is generated, this is also true for the so-called "prebiotic synthesis".

The greater the number of synthesized molecules, endless possibilities are generated, but this also implies a series of associated complications. First of all, the molecules that are part of living things are

very specific and belong to particular chemical groups (for example, sugars, lipids, amino acids). If the number of biomolecules used by living beings is limited, compared to all the possibilities that exist; this means that there was some mechanism(s) of molecule selection to explain, for example, why only 20 amino acids are in fact proteinogenic. In the present paper authors explore a concentration and selection mechanisms that could very likely have been present on primitive Earth, thermophoresis. This could be promoted by small heat flows through tiny cracks, and they mimic that by microfluidic devices. The authors also evaluated other reactions, such as amino acid dimerization and cytidine phosphorylation, both key reactions in prebiotic chemistry. It is noteworthy that they show that there is an important effect of fluxes by increasing the yield in both reactions.

A. Originality and significance: if not novel, please include reference

The data presented in the article are novel and present data of great relevance in the area of prebiotic chemistry that can lead to broad discussions and further work in the future.

B. Data & methodology: validity of approach, quality of data, quality of presentation

Microfluidic devices have only recently been used to perform prebiotic chemistry experiments, although their use in other fields is widespread. The working group is an expert in the design, implementation and adaptation of MDs, and its work is very detailed and careful. The analytical method that has been used is novel and allows to overcome some of the inconveniences of the use of devices, although it is still limited by the size of the sample and the sensitivity of the equipment used for quantification.

C. Appropriate use of statistics and treatment of uncertainties

The statistics were made with samples in triplicate, and some values show variations, but the treatment of the data is explained in detail.

D. Conclusions: robustness, validity, reliability

The conclusions are well founded and based on data.

E. Suggested improvements: experiments, data for possible revision.

It is not mandatory; however, it could be interesting to evaluate in the future the possible role of heat fluxes in enantiomeric selection.

F. References: appropriate credit to previous work?

The work is well referenced.

G. Clarity and context: lucidity of abstract/summary, appropriateness of abstract, introduction and conclusions

I really appreciate that the manuscript is well written and logically organized, so the reading is easy and

meaningful. I have punctual comments and questions that could help to improve the manuscript.

1. Figure 2. Setup and thermophoretic enrichment of prebiotic organics in a single heat flow chamber. I have problems with my eyesight, and it is difficult for me to distinguish between A and U in the Figure. Probably other grey tone could help to improve the resolution.

2. In the case of Figure 3, please include the name and the abbreviation of the AA in the Figure description, just the abbreviation, or just the name of the amino acid for homogeneity. For example. "Analysis using HPLC reveals a strong enrichment of aliphatic amino acids (I, V, L) at the bottom (orange shade) against G (79-84 %)" would be "Analysis using HPLC reveals a strong enrichment of aliphatic amino acids [isoleucine (I), valine (V) and leucine (L)], at the bottom (orange shade) against glycine (G) (79-84 %) and serine (S), asparagine (N) and glutamine (Q)". Or just "Analysis using HPLC reveals a strong enrichment of aliphatic amino acids (I, V, and L), at the bottom (orange shade) against G (79-84 %) and S, N and Q".

3. Extended Data Figure 1. Experiment setup and analysis. Please include what represents 1 in the figure description.

4. Regarding the enrichment of 3',5'-cyclic nucleotides in different solvents, there is a difference in selection depending on the solvent. While G and C are more abundant in formamide, A and U are slightly enriched in water. This is very attractive, but what are the implications of this finding? Since formamide has been proposed as a good choice for some prebiotic chemical reactions, it was probably mandatory for others?

5. Please check in the Methods section the subscript (H₂O) in LC measurements.

Further Questions

1. Whether thermophoresis can induce a separation (and accumulation) of molecules, even though there was no difference in your experiments of nucleoside selection, do you consider this process could explain a biased selection of the enantiomers of other molecules? Amino acids, for example. There is evidence that the eutectic point of a solvent (water) favours the amplification of the enantiomeric excess of amino acids (Klussmann et al. 2006).

Klussmann, M., Iwamura, H., Mathew, S. et al. Thermodynamic control of asymmetric amplification in amino acid catalysis. *Nature* 441, 621–623 (2006). <https://doi.org/10.1038/nature04780>

2. In addition, we do have just 20 proteinogenic α -amino acids and we still do not know what was the force that selected them. Is there any possibility for thermophoresis to have influenced the selection of those molecules?

3. Have you explored the effect of ionic strength on any of these processes? The interaction of fluids with minerals causes the dissolution of the rock and the consequent increase of ions in solution. This effect could greatly influence the behaviour of some organic molecules.

Author Rebuttals to Initial Comments:

Referees' comments:

Referee #1 (Remarks to the Author):

Mast and co-workers report on the separation of mixtures of prebiotically relevant building blocks through heat flows in confined volumes. This is very exciting work. Although I have several questions (see below), the work is impressive and potentially of very high relevance for the field of prebiotic chemistry, where no convincing, general methods exist that could yield highly enriched mixtures of starting materials for chemical reactions. This work demonstrates that such a method might indeed exist in the ubiquitous cracks in rocks.

We thank the reviewer for this initial positive assessment of our work.

My first main point of criticism is that I struggle with the presentation of the data. For example: page 4: In the top fraction, 2AI is depleted by 40 % compared to \bar{c}_{top} while 2-aminooxazole (2AO) is enriched by 40 %. Upon direct comparison, this corresponds to an enrichment of 2AO over 2AI by 142 % = $\bar{c}_{2AO,op}/\bar{c}_{2AI,top} - 1$ in the top fraction (Extended Data Fig. 2).

This is an important point raised by the reviewer. In the old manuscript, we had attempted to make the data easier to understand by presenting them in different ways. For example, the bar plots were intended to show the change in concentration at each chamber position relative to the concentration at that position averaged over all species, while the heat maps were intended to show pairwise enrichment. We believe, as does the Referee, that this is suboptimal. We now present all data uniformly as heat maps in which the pairwise enrichment of all species of a mixture at the lower (bottom) and upper (top) chamber positions is shown (see updated Figures 2 and 3). In addition, we now provide the raw data (Supplementary Tables 4-64) for all shown measurements which contains the integral values from the HPLC measurement, the respective resulting concentrations, the normalized values and enrichment values per experiment to allow maximum traceability.

Normalization of the data is necessary for a comparable presentation of the different data sets in the main manuscript, since we have different initial concentrations due to the variety of mixtures used. To make this clear in the manuscript, we now write in line 117 to 121 in the Results section:

"The concentration ratio of the respective substances in each section relative to the chamber-averaged concentration (= total concentration c_0) of this substance was then determined by HPLC. In this way, measurements of different mixtures of nucleobases and, e.g., of amino acids could be compared despite different initial concentrations of the individual components."

And in line 123 to 126 of the same section:

"All experiments were performed in triplicate, from which the mean enrichment $\overline{[A]_j/[B]_j} - 1$ between compounds A and B in the top ($j=top$) and bottom ($j=bot$) chamber parts was determined (see Methods, Fig. 2-3, Extended Data Fig. 2-5, explanation in Supplementary Table 4 for Fig. 2a, further raw data in Supplementary Tables 5-64)."

Normalization is then covered in more detail in the updated Methods section, "Treatment of data". We show here with Eq. 1 and 2 how first the concentration ratio of two species at one chamber position is calculated and how this is finally averaged over all three replicates. Since there are minor, setup-related temperature gradient variations (1-2K) between the repeats, which have the same effect on all species of the respective mixture, a calculation of the concentration ratio after averaging over all repeats would distort the enrichment value per experiment. To make this clear, we write in the methods section, line 664 to 669:

"The averaging over the triplicates is done only after the calculation of the concentration ratios. This is necessary because the temperature gradients between the replicate experiments differ slightly (1-2 K). This affects the concentrations of all species present in the respective mixture of a replicate equally (see Supplementary Figure 7), so that a calculation of the concentration ratio only after averaging the species concentrations would lead to a distortion of the enrichment value actually present in the heat flow chamber."

In Figure 2a of the main text, we further show an example of typical depletion and accumulation in the upper and lower chamber regions, respectively, using 2AI and 2AO to make clear how the enrichment is reversed at the top compared to the bottom.

- I might be missing something, but because the authors did not provide the actual concentrations measured, it is very difficult to check this calculation. By also not providing the numbers for the average concentrations in the top and bottom fractions, it is difficult to appreciate the magnitude of the effect. I am not suggesting the data is wrong, but the current presentation makes it (for me) impossible to follow.

We fully agree with the reviewer's comment. In Supplementary Tables 4-64 we now show all raw data, concentration values and enrichments with reference to the equations used from the Methods section.

- Why is the label in Figure 2a $c_0/2$ in the top, but $2c_0$ in the bottom fraction?

The bar graph plot in Figure 2a-c first calculates the concentration averaged over all species ($\langle c \rangle_{\text{top}}$, $\langle c \rangle_{\text{bot}}$) and then compares how the concentrations of individual species deviate from this average. Due to thermophoretic depletion, this mean value in the upper part of the chamber is well below the initial concentration c_0 ($\rightarrow \sim 0.5 * c_0$), while in the lower part of the chamber it is well above the initial concentration ($\rightarrow \sim 2 * c_0$). A disadvantage of this visualization is that it depends on which other species are included in the mixture.

As described above, in the main text we now move away from this suboptimal representation and map all the data in the form of the heat maps, which show the pairwise enrichment of all the species contained in the respective mixture. This representation does not depend on the other components of the respective mixture as long as the concentrations of the individual species remain small enough not to affect each other thermophoretically, as was the case in our study (aqueous solution, see e.g. doi.org/10.1073/pnas.060387310).

A second main concern is the reliability of the data. In Extended data Fig 2, the absolute error associated with the 142% is 52%. What does this mean? I would expect the absolute error to be the difference between the measured value and the actual value. But here it is normalized against the average concentration.

We agree with the reviewer that the presentation of the error was misleading. In the revised manuscript, we present the standard deviation as an absolute enrichment value, grouped in parentheses with the average value, e.g., (142 +/- 52)%. In the given example the enrichment value thus varies from (142-52)%=90% to (142+52)%=194%. To make this clear, we added to the figure captions of the EDF/SI figures containing the error maps, here exemplified EDF2a:

"The error maps each show the absolute enrichment error e.g. 2AI vs 2AO: (32±3.6)%, so the error range varies between (32-3.6)%=28.4% and (32+3.6)%=35.6%."

- In the same context, on page 5 the authors state: Thus, a single heat flow chamber can already build up significant relative concentration differences.". What do the authors mean with significant. It seems to me that the large error would indicate that statistical significance is often not reached?

To specify the significance of the respective results, we now also indicate the error/standard deviation for each enrichment value in the text. Accordingly, significance means that the error value is significantly smaller than the mean value. E.g., we write in line 212 to 217:

"Adenosine nucleotides are concentrated between (43 ± 1)% and (62 ± 0.3)% more than the adenosine nucleoside (Ado) in the lowermost fraction due to the additional phosphate group that increases the charge and thus the thermophoretic strength³⁵. It should be emphasized that 2',3'- and 3',5'-cyclic AMP are accumulated up to (14 ± 0.6)% more than linear 2',3'- and 5'-AMP. Despite being identical in mass, 2'-AMP is enriched by (4.1 ± 0.3)% over 3'- and 5'-AMP."

We also make clear the difference between significant and non-significant enrichments with appropriate examples and write in line 164 to 167:

"While thermophoretically similar AAs such as aspartic acid (D) and arginine (R) are not significantly separated from each other $\frac{[D]_{\text{bot}}}{[R]_{\text{bot}}} - 1 = (0.2 \pm 2.1)\%$, we found a massive separation of isoleucine (I) against glycine (G) by $\frac{[I]_{\text{bot}}}{[G]_{\text{bot}}} - 1 = (84 \pm 24)\%$ in the bottom fraction."

The concern about reliability/accuracy is especially important because the presumed prebiotic effect on the separation of species in rocks will be much larger than the separation effect seen in a single chamber. But all experiments are on single chambers and these values are then used as input for the simulations. The authors should provide realistic confidence intervals for the simulations.

The reviewer is absolutely correct in this assessment, which is why we report the corresponding standard deviation for all experimental enrichment values as described above (except for those in scatterplots, which contain the values of all replicate measurements). We now also give the standard deviations for the numerical simulations. For example, we write in the line 330 to 332:

"Under the same boundary conditions as before, 3',5'-AMP is concentrated to a maximum of (41 ± 16)-fold relative to adenosine. In other chambers, an (11 ± 1.9)-fold enrichment of adenosine versus 3',5'-AMP is achieved."

We have now added the procedure for calculating the numerical error from the experimental standard deviation in the methods part under "Modeling of a system of connected cracks/heat flow chamber" (line 745 to 805).

We would also like to mention that we now experimentally show the increase of the heat-flow-driven enrichment of compounds by a network of multiple heat flow chambers in the new Figure 4a-c.

- Accordingly, for 2'- versus 3'- and 5'-AMP, a maximal enrichment of 35 % is achieved in the lowest micrometer instead of the 4 % obtained by averaging over a quarter of the volume (Fig. 2d) This is very cryptic. Did the authors MEASURE this, or is it a result of a calculation (I appreciate that the bottom micrometer is most concentrated, but how do the authors get to these numbers?)

We agree with the Referee that this issue should be better explained. With this statement, we refer to the well-established experimental fact that thermogravitational chambers yield an exponential concentration profile after reaching the steady state (see, e.g., doi.org/10.1073/pnas.0609592104). If, as is the case with our measurement method, the concentration is averaged in 25% of the chamber volume by dividing it into four chamber parts, the result does not reflect the maximum possible concentration or enrichment at the lowest trap edge. As requested by the reviewer, we have cut the content of a sample chamber, being an exemplary mixture of amino acids also used for Figure 2b, into not only 4, but 12 volume fractions. We present the corresponding concentration profile in Extended Data Figure 2b, which shows a significantly higher concentration for the lowest 12th than for the lowest fourth, proving our point. The disadvantage is the correspondingly larger influence of small variations in the cutting of the frozen chamber contents and the associated larger error. We calculate

the corresponding exponential concentration profile and consequent enrichments as described in doi.org/10.1073/pnas.0609592104 and show it in Extended Data Figure 2c/d with the example of 2-aminoazoles. For better comprehensibility, we now elaborate on this in line 225 to 241:

"As shown above, a single heat flow chamber can already build up significant relative concentration differences. Since the concentration profile from thermogravitational accumulation is known to increase exponentially towards the bottom of the heat flow chamber^{45,46}, large enrichments at the very bottom of the chamber are averaged out by thawing of the individual quarter fractions. As a result, the maximal enrichment measured in the HPLC is lower than the maximum value expected in the bottom few millimeters of the heat flow chamber. To quantify this effect, we performed an additional experiment in which we divided the chamber content into 12 individual fractions. As expected, the lowest 1/12th of the volume, corresponding to the lowest four millimeters of the chamber, had a significantly higher concentration than the lowest quarter in a comparative experiment, but at the price of an increased error due to the now higher impact of volume variations when manually cutting the frozen chamber content (see Extended Data Fig. 2b). For longer accumulation times reaching steady state, the concentration profile is expected to show an exponential shape according to the Soret coefficients determined in this work and published literature^{45,46} (Extended Data Table 1, Methods Eq. 8), so that, e.g., a maximum enrichment of 2Al versus 2AO of 406 % (see Extended Data Figure 2c) is to be expected at the lowest chamber position in contrast to the (32 ± 3.6) % (Fig. 2a) in the lowest quarter after 18 h."

We also show how we calculate the profiles in the methods in line 739 to 744:

"Steady-state concentration profiles shown in Extended Data Fig. 2c/d and Supplementary Fig. 4 were obtained by calculating

$$(Eq. 8) \quad c_i(y) = \exp\left(-\frac{q_i}{1 + \frac{q_i}{10080}} S_{T,i} \Delta T \frac{y}{a}\right), \quad \text{with } q \equiv \frac{\Delta T \beta g \rho a^3}{6 \eta D_i}$$

where β denotes the volume expansion coefficient of water, a the distance between the hot and the cold side of the heat flow chamber, η the dynamic viscosity of water, and y the space coordinate along the height of the chamber."

-The data in Figure 3 (ignoring the difficulty I have discussed above with respect to interpreting the actual numbers) are very interesting, as they show that AAs separate strongly in these thermophoretic chambers. Have the authors attempted to separate all AAs in one single experiment? It would be very interesting to see if a similar degree of separation would be achieved, or that interactions between the AAs would diminish or enhance the separation. This would be a very important experiment to do! The prebiotic relevance of the work is to a large extent built on the need for purification of reagents prior to a reaction. Therefore, the reagents must be enriched from very complex mixtures (not pairwise...).

We absolutely agree with the reviewer, which is why the enrichments from the present Figure 2b were already before determined from a mixture of all indicated amino acids (not pairwise). We write accordingly in line 162 to 163:

"To determine their thermophoretic separation, we analogously investigated a mixture of all 20 proteinogenic AAs at physiological pH (Fig. 2b)."

In the methods section, we describe in detail the method by which we were able to measure the respective amino acids by HPLC (see Supplementary Figure 12 for exemplary chromatograms).

-The simulations in Figure 4 are very elegant and provide a realistic insight into how mixtures might be separated in rocks with multiple cracks, each with a temperature gradient. Can the authors discuss how relevant a value of 2K for the temperature gradient is? Would this be a lower estimate? I can well imagine that the temperature difference in a microporous rock is 1K or even just a fraction of a degree across a single crack?

We agree that it is important to also discuss smaller temperature gradients and the realism of the system shown here. We have therefore added the following text in Supplementary Discussion 1 in line 179 to 201:

"The exact determination of the temperature gradients occurring in natural systems, for example in fissures in basaltic glass, is difficult. Therefore, a discussion of the possible enrichments by heat flux driven chambers at very small temperature differences as well as the natural conditions, which also make larger temperature gradients appear realistic, is relevant. As shown in the main text in Figure 4f, even in small networks with very moderate temperature gradients of only 2 K, an enhancement of e.g. I vs G of (5.6 ± 0.5)-fold can already be achieved. And even for even smaller temperature differences, relevant enrichments are obtained. Thus, an enrichment of I vs G of (3.2 ± 0.2)-fold can still be achieved at 1 K and (1.9 ± 0.1)-fold at 0.5 K. As shown in Figure 4f, these values can be further boosted by the size of the system, but our calculations here are limited by the measurement accuracy of the Soret coefficients involved. Due to the accumulating error of the heat flux chambers connected in series, a calculation of systems N>30 is no longer reliably possible. However, we expect here a continuation of the behavior shown in Figure 4f and thus a compensation of the smaller temperature differences discussed here. Even larger temperature gradients are likely to be quite realistic within natural systems. Varying across orders of magnitude in fracture size and associated widely varying flow velocities, fast flows can serve as effective heat sources and sinks for narrower adjacent channels^{3,4}, implementing a wide range of possible temperature differences in which the effects shown can take place. Also, it should be noted here that the temperature gradients shown here, as discussed above, need not act all the time and could also fluctuate over the local dimension (for a stability analysis over local fluctuations, see reference 2). In summary, the systems discussed here may well be considered realistic and thus may have made a relevant contribution to prebiotic chemistry."

We also reference this text in the main body with an example value of 2K in line 317 to 321:

"Interestingly, even a slight temperature difference of only 2 K drives an up to (5.6 ± 0.5)-fold enrichment of I versus N in such a relatively small network (see Supplementary Discussion 1 for a detailed treatment). The

enrichment is further boosted with increasing network size, in particular for the thermophoretic weaker species (N vs I, Fig. 4f, lower).“

-Also: for the simulations, did the authors really use a flow rate in the range of 10^{-3} nL per s??

Fluid flow in geological systems as for instance connected fractures is highly complex and under investigation in the geoscience community since more than 50 years (without final conclusion). Permeability and thus fluid flow in fractures does depend strongly on the aperture of the fracture and also on the surface roughness of the fracture. The permeability of the explored geological systems, e.g. basaltic rocks can easily span 8-10 orders of magnitude, corresponding to a very broad range of fluid flow rates as well (doi.org/10.1029/2021RG000744, doi.org/10.1029/2009JB007047). We thank the reviewer for this question that helped us to better elucidate this complexity.

To account for this, the lower flux limit mentioned by the Referee was therefore set to model the realistic situation as closely as possible. As now shown in Supplementary Figure 8, in most of the heat flow chambers/cracks in our network system, one mostly finds a flow rate between 1-10nl/s.

My second main point of criticism concerns the (predicted) increase in conversions of chemical reactions in networks of cracks. I think these modeling results are not sound.

- First, for the reaction between cytidine and TMP, the model cannot use an equilibrium equation. This reaction is irreversible. Second, the dimerization of glycine is not a trimolecular reaction. Instead, first one glycine is activated via phosphorylation with TMP and then the second glycine reacts with the activated intermediate. Thirdly, I don't think one can/should use the product-dependent reaction rate, as there is no 'product inhibition'.

We agree with the Referee that the reaction model used was an oversimplification for describing glycine dimerization. Therefore, we now have a reaction system as in Eq. #9 to #13, which follows the reaction scheme as suggested by the Referee as shown, for example, in [doi.org/10.1016/S0040-4020\(01\)90868-3](https://doi.org/10.1016/S0040-4020(01)90868-3) that describes the TMP induced peptide synthesis. Accordingly, this reaction is irreversible and instead of a trimolecular reaction it is a sequence of reactions: First, a glycine is activated by TMP (Eq. #9) and finally dimerized by reaction with another glycine (Eq. #11). Further, there is no product inhibition, but degradation reactions (Eq. #10, #12, and #13) in which the intermediate/end products break down again by hydrolysis. Also, considering the comment of Referee 2, who saw the chemical reaction only as an add-on, and due to the fact that heat flux-driven phosphorylation of nucleosides has been shown before in doi.org/10.1038/s41557-019-0299-5 (though there not in an aqueous system, but driven by surface fluxes at a gas/water interface), we focus entirely on glycine dimerization in the revised manuscript. To determine the reaction rates described in Equations #9 through #13, we performed a simultaneous rate fit for all bulk glycine dimerization experiments started with different initial glycine and TMP concentrations and measured at different time points. As shown in Figure 5c, the model given in Eq 9-13 fits the experimental results very well. In the text we describe this in line 380 to 385 (details in the Methods section):

“Therefore we studied the dimerization of glycine experimentally in bulk under TMP titration at a temperature of 90 °C and with initial pH 10.5, and after multiple time points up to 120 h in bulk solution, respectively (see Methods, Fig. 5c). We could then simultaneously fit the reaction rates of the model shown in Eqs. 9-13 to these results, which allowed us to quantify how a network of heat flow chambers boosts reaction yields.”

Using these reaction rates, we were then able to determine the reaction yields in the network of heat flow chambers.

- Furthermore, considering the long reaction times, I wonder whether (or rather, suggest that) hydrolysis reactions should be taken into account. Both TMP and the phosphorylated and dipeptide products can hydrolyze under basic conditions. It might well be that any increase in yield would be eliminated by hydrolysis.

We also took into account the hydrolysis of the reaction. We write accordingly in the main text, line 386 to 393:

“The network was implemented in silico by 20 by 20 connected heat flow chambers, fed with a mixture of 1 μ M TMP and glycine at a flow rate of 1 nl/s per inlet (Fig. 5d). The network continuously provides the reactants according to the accumulation characteristics shown in Figure 4, taking into account the high Soret coefficient of TMP $S_{T,TMP} \sim 7 \cdot 10^{-3} (1/K)$. As shown in Fig. 5c, right, the maximum reaction yield is already reached after 16 h, but the relaxation time of a chamber with through-flow in the range $Q_{flow} = 1 - 10 \text{ nl/s}$ is expected to be in the order of $Volume_{chamber}/Q_{flow} \sim 10^1 - 10^2 \text{ h}$. Therefore, in order to sufficiently account for the hydrolysis included in the model, we calculate the product yield after 120 h reaction time.”

As it stands, Figure 5 should be backed up by experimental data. Now, the reactions are carried out in bulk, at least an experiment in a single trap should be done to see if there is any acceleration at the top or bottom of the trap.

In accordance with the Referee's request, we performed glycine dimerization in a single heat flow chamber in the revised manuscript (see new Figure 5b). As clearly seen there, thermal non-equilibrium can raise the reaction yield from an undetectable level to over 3%. Due to the fact that the concentration of the reactants is strongly dependent on the temperature gradient used and this can vary by 1-2K over the different repeats due to setup, a relatively large measurement error as shown here, but this was to be expected. Nevertheless, the measured reaction yields are significant (especially at the bottommost chamber position at 10mM glycine starting concentration (see Fig. 5b)). Controls were run for both the respective maximum temperature occurring in the heat flow chamber to avoid a false positive result in which the reaction is merely boosted by the higher temperatures in the heat flow chamber. We write accordingly in the text from line 371 to 376:

"To determine how glycine dimerization would benefit from this selective enrichment, we filled a heat flux chamber as described above with a mixture of 1 mM (Fig. 5b, pink) or 10 mM glycine (Fig. 5b, purple), and 1 mM TMP, respectively. After 16 h run time with a temperature difference of $\Delta T = 14\text{ K}$ the product yields were increased from an undetectable level to $(3.6 \pm 0.6)\%$ compared to the isothermal controls ($\Delta T = 0$, black) in heat flow chambers at the upper experimental temperature ($T = 85\text{ }^\circ\text{C}$)."

We thank the reviewer for the critical comments. The experiments proposed and now performed clearly show that prebiotically relevant reactions can be enhanced by heat flux driven selective enrichment. We think that we could thus address all suggestions of the Referee.

Referee #2 (Remarks to the Author):

The manuscript by Matreux and colleagues discusses how prebiotically relevant molecules could be separated and enriched from complex mixtures using temperature gradients that might occur in rock fractures. The authors used a microfluidic heat flow chamber to study the accumulation of molecules from different mixtures within this simulated rock fracture. HPLC analysis by UV and fluorescence detection was used to determine concentrations within different sections of the flow chamber. From the thermophoretic properties the accumulation and separation of complex mixtures in a network of interconnected "rock fractures" was modelled. The authors show how the accumulation of molecules might be beneficial for prebiotic reactions to boost the yields.

The authors identified an extremely important problem in prebiotic chemistry, as most reactions provide product mixtures. Side products are often not compatible with downstream reaction steps and therefore intermediates need to be purified or enriched. The work in this manuscript provides an innovative concept to potentially solve this problem.

We thank the Referee for the assessment of the importance of this problem and its solution, which we also share.

Having said this, the work presents a good starting point but I am not yet convinced that the authors fully solved the problem. The theoretical modelling calculates a network of interconnected rock fractures, which also assumes a flow through the rock cracks. However, the authors use a fully closed heat flow chamber for experimental validation. In previous work the authors have already shown that concentration gradients can be achieved in such a system. They now transfer this concept to a much larger set of molecules and show that there are differences in the concentration gradients between the molecules. This leads to relative enrichment/depletion of compounds within a defined section of the flow cell.

However, interconnected rock fractures also means that compounds need to be transferred from one rock fracture to the next, which has not been demonstrated here. Such a transfer might interfere with the convection inside the heat flow chamber, which is responsible for the concentration gradient of the tested molecules. Therefore, to validate the theoretical concept, the authors need to show how they can connect flow chambers together in order to transfer compounds by flow. This means the system is actually open with input and output as shown in Figure 1b. Here it would be important to show that connecting flow chambers can potentiate the enrichment without losing too much material. Experimental demonstration of compound separation by interconnecting several flow chambers would be enough to merit publication in nature.

We agree with Referee that experimental proof of the concept of multiple interconnected heat flow chambers would make the model shown much more convincing. For this reason, we experimentally created a minimal network of 3 connected heat flow chambers and fed it with a continuous flow rate of 1 nl/s (the same used in the numerical model) with a mixture of amino acids, which was also used for Figure 2b. The result is shown in Figure 4a-c. Accordingly, we write in the text in line 245 to 268:

"To mimic such a complex system of interconnected chambers, we first performed a proof-of-principle experiment (Fig. 4a-c). Here, we applied a flow of 1 nl/s of a mixture of compounds to the first heat flux chamber "1" (violet shade, Fig. 2a), which is then branching to two chambers "2" and "3" (brown and blue shade, Fig. 2a). Understanding the enhanced enrichment by these three chambers would then enable us to model even larger systems numerically. In the experiment shown in Fig. 4a, we used the same mixture of amino acids as in Fig. 2b and applied a temperature difference of 17 K. To determine the spatially resolved enrichment, we froze the chambers after a run time of 60 h. We then divided them into the color-graded areas shown in Fig. 4a for analysis via HPLC. Fig. 4b shows the position and concentration of exemplarily selected amino acids F, I, and N with the corresponding color assignment (see Extended Data Fig. 6c for A, C, K, L, P, S, T, and V). The scatter plots including I and N clearly show the separation of concentration ratios per chamber. Figure 4b/c displays all three repeats. The enrichments in chamber "2" (brown shade, $[N]/[I] = 19$ -fold) and chamber "3" (blue shade, $[I]/[N] = 3.6$ -fold) are higher than those achievable in chamber "1" alone (violet shade) (Fig. 4b, left). This shows that the applied through-flow is not only compatible with obtaining enrichments in a single chamber, but that the interaction of three chambers in a network enhances the possible separations substantially. As expected, the maximum enrichments for thermophoretically different amino acids, such as I versus N are significantly greater than those for thermophoretically similar amino acids, such as I versus F (Fig. 4b/right), see more examples in Extended Data Fig. 6c. There, we found a maximum enrichment of $[F]/[I] = 2.6$ -fold (chamber "2", upper position) and $[I]/[F] = 1.6$ -fold (chamber "3", lower position). As expected, the enhancement effect becomes stronger for larger temperature differences (Fig. 4c, blue $\Delta T = 12$ K, red $\Delta T = 17$ K) and disappears without a heat flux (green, $\Delta T = 0$ K)."

As shown in Figure 4b, the interconnected chambers are not only compatible with the flow rate used, but also allow for significantly increased enrichment over a single chamber. In addition, the scatterplot shown in the numerical model (Figure 4e) is consistent with the experimental results, but is significantly more expansive due to the larger network used (not just three, but 20x20 chambers). Therefore, the experiments provide the necessary extended foundation to explore larger networks as shown in Figure 4d-i.

The demonstration of the "prebiotic chemistry" is not really needed and feels more like an add on.

To balance with the comments between Referee 1 and Referee 2, we have now used only glycine dimerization as an example but also incorporated in a heat flow chamber as suggested by Referee 1, also because phosphorylation of nucleosides by heat fluxes has been shown before (but there at water-air interfaces and without TMP) – see

doi.org/10.1038/s41557-019-0299-5 for reference. Moreover, we could now model the reaction with kinetic rate equations and thus give a proper theoretical explanation of the reaction performed in more extensive crack networks.

I might have fully misunderstood some of the figures but it seems the provided data in this manuscript is not always consistent or missing. The data in Figure 2c and 2d does not match. In 2c (top) adenosine is actually depleted in comparison to the average concentration. In contrast, the investigated phosphorylated adenosine species (AMPs) are all enriched relative to the average concentration. Figure 2d (top), however, shows the opposite (adenosine being enriched compared to the different AMPs). A similar inconsistency is found in extended data figure 4f (bot), which states that all phosphorylated species (CMPs) are enriched in comparison with cytidine by at least 18%. A look at figure 2d, shows that only 2',3'-CMP is more strongly enriched compared to cytidine.

We agree with the Referee that the different way of presentation in old Fig 2b/c compared to Fig 2d at first sight seems inconsistent and contradictory. The reason for this is that in Fig 2b/c the respective species concentrations are compared with the mean concentration averaged over all species of the respective pool of molecules (e.g. in the case of adenosine all ribonucleosides) at the respective position, whereas in Fig 2d (and for other heatmaps of the later Figures) the pairwise enrichments between 2 compounds in a mixture were shown (as defined in Eq. 1 and 2). Due to the averaging over all species in Figure 2b/c, the indicated deviation from this mean value depends on the other species present in the mixture. Thus, in the example shown by the Referee in Fig2c, a mixture of the different nucleosides was used in the first data group (Nucleosides), while a mixture of adenosine nucleotides (with different phosphorylation state) and adenosine was used in Figure 2d. Since nucleotides have a much stronger thermophoresis than the nucleoside due to their additional phosphate group (and therefore charge), they are concentrated more strongly in the lower chamber region. Therefore, more adenosine remains in the upper chamber region, which is why it is enriched there compared to the other species. In the mixture used in Figure 2c, the other nucleoside species are present instead of the nucleotides, with respect to which the adenosine is depleted in the upper chamber region.

We therefore decided to present all data in the revised version of the manuscript in the form of the heat maps (previously shown in the old Figure 2d), in which the enrichments of one species are shown relative to another and no changes relative to a species-averaged concentration occur anymore. We want to refer for more details to the answer to Referee 1.

Some of these enrichment blots suddenly include 2'-phosphorylated nucleotides but the data is nowhere to be found.

For all data shown in the main text and Extended Data Figures, we now append in Supplementary Tables 4 to 64 the raw data containing HPLC integrals, concentrations and enrichments as described in the methods.

In addition, the main text states "The most significant enrichment of guanine species in the bottom fraction is achieved in mixtures of nucleosides or 2',3'- phosphorylated nucleotides". Looking at figure 2c shows that only guanosine-5'-monophosphate is slightly enriched. All other guanine species, including the nucleoside, are depleted.

We thank the Referee for this comment, in fact, this is a typo on our part. Actually, the cytosine species were meant here, which are enriched in mixtures of nucleosides and 2',3'-phosphorylated species in the lower chamber region. As part of the changes to this manuscript and the more meaningful pairwise presentation of the enrichments, this text has been removed.

Given these inconsistencies figure 3 cannot be properly evaluated because the source data is missing. All source data (integrals from HPLC analysis, average concentrations, enrichment in % etc. should be provided for example in a table in the extended data). A representative example chromatogram of the experiments should be provided as well to check peak separation or relative peak areas.

We now report the source data in Supplementary Table 4 to 64, which contain the integral data from the HPLC measurements, as well as the determined concentrations and consequent enrichments. We also show, by way of example, in Supplementary Figure 12 the peak separations achieved from the HPLC measurements for the mixtures used for the Figures in the main text.

Other comments:

- The authors usually compare the same class of molecules for enrichment. In extended data figure 4 they tested chirality as well. It would be interesting to also test sugar/nucleoside isomers (e.g. ribose vs. other pentoses or alpha vs. beta nucleosides). Separation of such compounds could have a huge impact.

We agree with the Referee that especially the thermophoretic separation of prebiotic components with different chirality would be very interesting. This was also raised by Referee 3, who had requested chiral selection of amino acids. In order to balance the additional experiments performed at the request of the referees, we therefore performed further enrichment experiments with amino acids Serine and Threonine of different chirality (see Extended Data Figure 5t and 5u). Due to the known diverse side reactions of the aforementioned sugars, an enrichment study for chirality selection of different sugars is beyond the scope of this work, but could be indeed interesting to study in the future.

- The authors state that "...the enrichment measured in the HPLC is significantly lower than present in the heat flow chamber." They assume this because they take the lower 25% fraction. To validate this claim, the authors could also use the lower 10% fraction. The lower 10% should have the largest concentration and therefore should be suitable for HPLC analysis. The concentration can then be calculated from a calibration curve and compared to the 25% sample.

We agree with Referee that the statement made in the old manuscript about the higher possible enrichments in the lowest chamber volume (e.g., the lowest 10%) should be verified with an experiment. As written in the reply for Referee 1, this was shown exemplarily in the new manuscript version on a solution of amino acids: Here, we have divided the chamber contents not into 4, but into 12 partitions and determined the concentrations in each case (see Extended Data Figure 2b). Due to the

smaller volumes per partition, the respective error increases while the positioning accuracy of the knife remains the same, which is why the division into 4 partial volumes is a good compromise.

- "Accordingly, for 2'- versus 3'- and 5'-AMP, a maximal enrichment of 35% is achieved in the lowest micrometer instead of the 4% obtained by averaging over a quarter of the volume (Fig. 2d)." It is not clear where the 35% are coming from but this value does not appear in Figure 2d.

We thank the referee for this advice. With this statement we wanted to point out that in our experiments we underestimated the actual maximum accumulation at the lowest end of the chamber by dividing the chamber volume into quarters and the resulting averaging. There, due to the well known exponential concentration profile (doi.org/10.1073/pnas.0609592104) forming in steady state, significantly higher concentrations and thus also concentration differences between the species would be expected. We would like to refer here to an answer above to a corresponding comment of Referee 1.

With the experiments shown for Extended Data Figure 2b, we were able to show that there is indeed a larger actual accumulation in smaller subvolumes and estimate this value by means of the theory used in doi.org/10.1073/pnas.0609592104 for the equilibrium case. We show the theoretical concentration profile at equilibrium in Extended Data Figures 2c and d using aminoazoles as an example and write accordingly in the text from line 225 to 241:

"As shown above, a single heat flow chamber can already build up significant relative concentration differences. Since the concentration profile from thermogravitational accumulation is known to increase exponentially towards the bottom of the heat flow chamber^{45,46}, large enrichments at the very bottom of the chamber are averaged out by thawing of the individual quarter fractions. As a result, the maximal enrichment measured in the HPLC is lower than the maximum value expected in the bottom few millimeters of the heat flow chamber. To quantify this effect, we performed an additional experiment in which we divided the chamber content into 12 individual fractions. As expected, the lowest 1/12th of the volume, corresponding to the lowest four millimeters of the chamber, had a significantly higher concentration than the lowest quarter in a comparative experiment, but at the price of an increased error due to the now higher impact of volume variations when manually cutting the frozen chamber content (see Extended Data Fig. 2b). For longer accumulation times reaching steady state, the concentration profile is expected to show an exponential shape according to the Soret coefficients determined in this work and published literature^{45,46} (Extended Data Table 1, Methods Eq. 8), so that, e.g., a maximum enrichment of 2AI versus 2AO of 406 % (see Extended Data Figure 2c) is to be expected at the lowest chamber position in contrast to the (32 ± 3.6) % (Fig. 2a) in the lowest quarter after 18 h."

- Methods: Treatment of data. The c in the equations is not defined but it seems this is the concentration? How the authors convert the HPLC peak integrals into concentration is not clear.

We thank the Referee for this remark. In fact, "c" meant concentrations, which was not defined in the old manuscript until far too late. In Supplementary Tables 4 to 64 we now give all raw data, from the HPLC integral values to the concentrations to the enrichments, and also indicate which equations were used in the method section for the respective conversions. The concentrations in the equations are now defined at the right time in the text. E.g., In line 650 to 654 we write:

"The normalized concentration $[A]_{j,k}$ per individual measurement was calculated for species A for fraction $j \in \{1,2,3,4\}$ and replicate number $k \in \{1,2,3\}$.

$$(Eq. 1) \quad [A]_{j,k} = \frac{[A]_{j,k,HPLC}}{\frac{1}{4} \sum_j [A]_{j,k,HPLC}}$$

with the HPLC-measured concentration $[A]_{j,k,HPLC}$ obtained from the integral values presented in Supplementary Table 4-64."

We thank the Referee for the helpful notes and comments. Following these, the new experiments of a minimal network of heat flow chambers show how the enrichment of prebiotic components from complex mixtures in a chamber can be further boosted. Thanks to these new results, we believe to have answered all the comments of the Referee.

Referee #3 (Remarks to the Author):

Heat flows purify and boost reactivity of >50 building blocks of life

Dear Editor, dear authors, I really enjoyed reading your paper that represents a good approximation to the study of complex processes in prebiotic chemistry studies with a novel approach. Overview and general recommendation

In prebiotic chemistry studies, the synthesis of many basic molecules for living beings has been achieved with great success. However, the synthesis is only the first step in the complex process that led to the emergence of the first organisms. When any chemical synthesis is carried out, an enormous variety of products is generated, this is also true for the so-called "prebiotic synthesis".

The greater the number of synthesized molecules, endless possibilities are generated, but this also implies a series of associated complications. First of all, the molecules that are part of living things are very specific and belong to particular chemical groups (for example, sugars, lipids, amino acids). If the number of biomolecules used by living beings is limited, compared to all the possibilities that exist; this means that there was some mechanism(s) of molecule selection to explain, for example, why only 20 amino acids are in fact proteinogenic. In the present paper authors explore a concentration and selection mechanisms that could very likely have been present on primitive Earth, thermophoresis. This could be promoted by small heat flows through tiny cracks, and they mimic that by microfluidic devices.

The authors also evaluated other reactions, such as amino acid dimerization and cytidine phosphorylation, both key reactions in prebiotic chemistry. It is noteworthy that they show that there is an important effect of fluxes by increasing the yield in both reactions.

A. Originality and significance: if not novel, please include reference

The data presented in the article are novel and present data of great relevance in the area of prebiotic chemistry that can lead to broad discussions and further work in the future.

We thank the Referee for the positive assessment of this work in terms of relevance to prebiotic chemistry.

B. Data & methodology: validity of approach, quality of data, quality of presentation

Microfluidic devices have only recently been used to perform prebiotic chemistry experiments, although their use in other fields is widespread. The working group is an expert in the design, implementation and adaptation of MDs, and its work is very detailed and careful. The analytical method that has been used is novel and allows to overcome some of the inconveniences of the use of devices, although it is still limited by the size of the sample and the sensitivity of the equipment used for quantification.

C. Appropriate use of statistics and treatment of uncertainties

The statistics were made with samples in triplicate, and some values show variations, but the treatment of the data is explained in detail.

As written in response to a previous Referee comment, we have now also added end-to-end error handling to the numerical modeling to make the data treatment and statistics more complete.

D. Conclusions: robustness, validity, reliability

The conclusions are well funded and based on data.

E. Suggested improvements: experiments, data for possible revision.

It is not mandatory; however, it could be interesting to evaluate in the future the possible role of heat fluxes in enantiomeric selection.

As suggested by the Referee, we performed additional enrichment experiments for mixtures of amino acids of different chirality (see below for a detailed description).

F. References: appropriate credit to previous work?

The work is well referenced.

G. Clarity and context: lucidity of abstract/summary, appropriateness of abstract, introduction and conclusions

I really appreciate that the manuscript is well written and logically organized, so the reading is easy and meaningful. I have punctual comments and questions that could help to improve the manuscript.

1. Figure 2. Setup and thermophoretic enrichment of prebiotic organics in a single heat flow chamber.

I have problems with my eyesight, and it is difficult for me to distinguish between A and U in the Figure. Probably other grey tone could help to improve the resolution.

Due to the now unified presentation of the data in the form of heat maps, we avoid the closeness of color. Each heat map shows the enrichment between a pair of components from a mixture rather than the enrichment against the pool average over all species (see responses to referee comments above). The data shown before in Fig. 2 is now presented in this unified way in Figure 3 and Extended Data Fig. 5. Species naming (along the heatmap axes) is now in black font for maximum contrast and ease of reading.

2. In the case of Figure 3, please include the name and the abbreviation of the AA in the Figure description, just the abbreviation, or just the name of the amino acid for homogeneity. For example. "Analysis using HPLC reveals a strong enrichment of aliphatic amino acids (I, V, L) at the bottom (orange shade) against G (79-84 %)" would be "Analysis using HPLC reveals a strong enrichment of aliphatic amino acids [isoleucine (I), valine (V) and leucine (L)], at the bottom (orange shade) against glycine (G) (79-84 %) and serine (S), asparagine (N) and glutamine (Q)". Or just "Analysis using HPLC reveals a strong enrichment of aliphatic amino acids (I, V, and L), at the bottom (orange shade) against G (79-84 %) and S, N and Q".

We thank the Referee for this remark. Indeed, the previous figure description was incomplete in this respect. Following the Referee's request, we now write in the caption of Figure 2b (which now shows the enrichment of amino acids):

"Enrichment in a mixture of all proteogenic amino acids (30 μ M each) in a single heat flow chamber using the same visualization as in (a). Analysis with HPLC reveals a strong enrichment of aliphatic amino acids isoleucine (I), valine (V), and leucine (L) at the bottom (orange shade) against glycine (G) (up to 84 ± 24 %) and serine (S), asparagine (N), and glutamine (Q) (up to 64 ± 17 %). Consistently, the aliphatic amino acids are strongly depleted in the upper part of the chamber (blue shade), resulting in an up to (287 ± 127) % higher local glycine concentration. See Extended Data Fig. 3-4 for measurements at other initial pH values and salt concentrations, and error maps."

3. Extended Data Figure 1. Experiment setup and analysis. Please include what represents 1 in the figure description.

In fact, we had not defined the part "1" of the setup and thank the Referee for this hint. Like part "3", this is a thermally conductive graphite foil which gives us high precision in controlling the temperature on both the front and back of the chamber. For clarity, we have added a description in the caption of Extended Data Figure 1:

"The sapphire-FEP-sapphire block is then placed on an aluminum base (2) covered by a heat conducting foil on the back (1) and front (3) for optimal heat conduction to the cryostat and the sapphire, and held in place by a steel frame (7)."

4. Regarding the enrichment of 3',5'-cyclic nucleotides in different solvents, there is a difference in selection depending on the solvent. While G and C are more abundant in formamide, A and U are slightly enriched in water. This is very attractive, but what are the implications of this finding? Since formamide has been proposed as a good choice for some prebiotic chemical reactions, it was probably mandatory for others?

We agree with Referee that this is a fascinating feature. In the new unified representation (see responses to Referee comments above), these data are now also shown as heat maps in which, analogous to the previous bar-plot representation in formamides, the nucleotides C (bottom) and G (top) are enriched (Extended Data Figure 5p), while in water the nucleotides U (bottom) and A (top) are enriched (Extended Data Figure 5o). This shows that changes in the solvent can have a complex influence on the selection of certain prebiotic components.

The implications of this ultimately depend on the reaction coupled to it - in this case, for example, this could lead to a slight shift in sequence bias in a hypothetical polymerization reaction. Without having actually carried out this reaction in situ, we did not want to speculate too widely on this in the main text, specifically also because the enrichments at base and nucleoside levels are significantly larger (see main text, Figure 3).

5. Please check in the Methods section the subscript (H₂O) in LC measurements.

We now write the 2 in H₂O correctly in subscript. We thank the Referee for pointing this out.

Further Questions

1. Whether thermophoresis can induce a separation (and accumulation) of molecules, even though there was no difference in your experiments of nucleoside selection, do you consider this process could explain a biased selection of the enantiomers of other molecules? Amino acids, for example. There is evidence that the eutectic point of a solvent (water) favours the amplification of the enantiomeric excess of amino acids (Klussmann et al. 2006). Klussmann, M., Iwamura, H., Mathew, S. et al. Thermodynamic control of asymmetric amplification in amino acid catalysis. Nature 441, 621–623 (2006). <https://doi.org/10.1038/nature04780>

Inspired by this question of the Referee, we additionally tested a mixture of amino acids with different chirality, respectively, for threonine and serine, in a heat flow chamber and determined the resulting enrichment (see Extended Data Figure 5t and 5u). We could not detect a significant separation of the different enantiomers and, therefore, see no indication that thermophoresis could at least directly contribute to symmetry breaking towards an enantiomer. We thank the Referee for the reference, which we now cite accordingly in line 181-182:

"We also found no thermophoretic separation between amino acids⁴⁸ and nucleosides of different chirality (Extended Data Fig. 5s-u)."

2. In addition, we do have just 20 proteinogenic α -amino acids and we still do not know what was the force that selected them. Is there any possibility for thermophoresis to have influenced the selection of those molecules?

We thank the Referee for this suggestion. Following this, we chose a mixture of 28 exemplary proteogenic and non-proteogenic amino acids and determined their heat flux-driven enrichment, as shown in Extended Data Figure 2g, f. For this purpose, we divided the chamber volume into 25% and 8% parts according to the previous referee comments in order to obtain a higher resolution of the enrichments. We did not find any significant enrichment of proteogenic over non-proteogenic amino acids. Accordingly, we write in the caption of Extended Data Figure 2g, f:

"(f) In a mixture of non-proteogenic and proteogenic amino acids and small organics molecules (diluted to 50 μ M and containing 0.02-fold lithium citrate, 0.01% phenol and 0.2% thiodiglycol), no clear pattern of separation is observed with non-proteogenic amino acids both on the strongly and weakly accumulating sides. Error maps defined as in (a). (g) The same is the case for extraction in 8 % fractions, with higher absolute values but also increased errors. Error maps defined as in (a)."

And in the main text line 179 to 181:

"When using a mixture of 17 proteogenic and 9 non-proteogenic amino acids, we found no clear bias towards either (Extended Data Fig. 2)."

However, this does not exclude that thermophoretically driven processes could be involved in such a selection. Thus, interactions of amino acids with other prebiotic components could form complexes that can be sufficiently selected by thermophoresis due to their size, structure or charge. A detailed investigation of such more complex systems will be part of future studies, however, we think this is outside the scope of this manuscript.

3. Have you explored the effect of ionic strength on any of these processes? The interaction of fluids with minerals causes the dissolution of the rock and the consequent increase of ions in solution. This effect could greatly influence the behaviour of some organic molecules.

To check the influence of different salt concentrations mentioned by the Referee, we now measured the solution of amino acids also used in Figure 2b over 3 orders of magnitude of NaCl concentrations (1mM, 10mM, 100mM) as an example. As shown in Extended Data Figure 4, there was no major influence of ionic strength at least in this concentration range. In fact, it is well known that thermophoresis of charged polymers or ions generally decreases with ionic strength (doi.org/10.1073/pnas.0603873103). However, under the thermogravitational accumulation studies in this work, the situation is much more complex since the thermophoretic concentration effect is enhanced by the co-occurring fluid convection, leading also to considerable salt concentration gradients (doi.org/10.1038/s41557-021-00772-5). Thus, areas of reduced salt concentration and thus restored thermophoresis of the remaining solutes also occur in the chamber volume. We write accordingly in the caption of Extended Data Figure 4:

" Pairwise enrichment between proteinogenic amino acids (30 μ M each) with different concentrations of NaCl. For all salt concentrations, analysis reveals a strong enrichment of aliphatic amino acids (I,V,L) at the bottom chamber against G (17-33 %) and serine, asparagine and glutamine (13-28 %). In the top fraction, the situation is inverted with up to 50+ %."

The same is true for enrichment of nucleobases, where we did not find major influence of ionic strength in 100mM phosphate buffer (Extended Data Fig. 5c).

And in the main text, we write in line 177:

"Different ionic strengths yield similar separation patterns (Extended Data Fig. 4)."

The enrichment by heat flux is thus stable for the salt concentrations commonly encountered by leaching as shown in doi.org/10.1038/s41557-021-00772-5.

We thank the Referee for the helpful comments and suggestions. In particular, the newly added experiments on the enrichment of proteinogenic versus non-proteinogenic amino acids or of enantiomers of amino acids have significantly broadened the scope of the manuscript.

Reviewer Reports on the First Revision:

Referees' comments:

Referee #1 (Remarks to the Author):

I would like to thank the authors for their responses to my questions. The new way of presenting the data (and the actual inclusion of the data) makes it much easier to appreciate these important and highly original results. I don't have any further questions. This work really provides an exciting new scenario for the emergence of complex molecules in a prebiotic setting by constructing complex synthetic routes that combine chemical reactions with purification by rocks.

Referee #2 (Remarks to the Author):

The revised version of the manuscript includes now a connected flow chamber experiment, dimerization of glycine in a flow cell and separation of non-proteogenic amino acids as well as separation at different ionic strength. In addition, the authors provided the source data for a better understanding of their results.

In the first round of review, I suggested a validation experiment of connected flow cells, which is now shown in figure 4. However, the data does not seem robust enough to substantiate the main conclusions. I assume that there might be a problem with their quantification methodology (as outlined below).

1) Technically it's great that the authors run 3 repeat experiments (supplementary table 14). However, I don't understand why the authors divide the flow cells into different number of sections in each repeat (ranging from 1 to 7). Therefore, a section in one repeat is not comparable with the section of the other repeats. Why they don't keep it consistent for full comparability?

2) There are large concentration differences between the repeats (sometimes up to 5-10x), but as outlined in 1) a perfect comparison cannot be made. According to the additional note, the authors attribute this to "small perturbances of the flow speed" or "generation of gas bubbles". If the authors are aware of those effects, they could have divided the flow cells into 4 instead of up to 7 sections (as they have done in Figure 2) to average out such effects. In addition, the authors could run the experiment at lower temperature to suppress the formation of gas bubbles. I couldn't find the actual temperature in the methods section that was used for the heating elements (top/bottom). Most importantly, I still believe that there is a potential problem with quantification (see also 3 to 5). The authors measured calibration curves in the range of 5-75 μM . All extreme values (supplementary table 14) seem to be outside this linear range and therefore might not be reliably quantified. As an example the authors measured two calibration curves for glycine and the equations are completely different (since they cover different ranges).

3) The authors ran a control experiment with $dT = 0$ (Supplementary Table 14), which shows that a

gradient is not formed. However, in this experiment a single amino acid (P) is suddenly selectively depleted in repeat 1. Repeat 2 shows no significant anomalies. A look at repeat 3 shows average concentrations that are about 10x lower.

In addition, the dT = 12 experiment (Supplementary table 14) shows selective depletion of C. As the authors explained in their additional note even gas bubbles can't explain a selective depletion. Together, this points towards a potential problem with their quantification methodology (see also 4 and 5).

4) The authors used the same amino acid concentrations (30uM) for the experiments shown in extended data figure 3 and 4. Repeat 1 in extended data figure 3 (Supplementary table 28) shows average concentrations between 13 and 34 uM (difference of up to 2.6x). A direct comparison of the average concentrations between extended figure 3 and 4 (Supplementary table 28 vs. 34) reveals that the values are about 3x higher for the latter (80 uM). Given that a closed flow cell and a starting concentration of 30uM was used, those values can't be correct. Of course, if there is a systematic error this might not affect the relative values but to be sure the authors should carefully check their data and quantification.

5) Further anomalies are for example found in Supplementary Table 4 (repeat 2) where the integral of amino acid Y suddenly increases by about 20x. In repeat 3 the same amino acid gives the following integrals for section top (26,892) and II (20,093). However, the concentration given in the table is 11.807 and 25.894 uM respectively. This can't be correct.

In addition, extended table 28 repeat 1 amino acid Y (section top) has an integral of 8,593 which corresponds to a concentration of 17.996. There is only one calibration curve of Y and the sample was divided into 4 sections (as in supplementary table 4). Given the linear relation of the calibration curve, something doesn't add up here.

Overall, I cannot check every single value but points 2-5 are in my opinion enough to question the reliability of the data. Since the main conclusions are based on quantifications it is important that the authors carefully check their data.

Other comments:

-Why the authors jump from a closed single flow cell directly to a complex flow-connected system of three flow cells? Ideally, they would first check how the flow influences the concentration gradient in a single cell and then move to a system of interconnected flow cells. Given the presence of secondary effects ("gas bubbles" or "perturbances of the flow speed"), this would be even more important to better understand the contribution of those factors in case there is indeed no way to suppress them (see point 2).

- in the methods section the authors state that they divide the flow cells shown in figure 4a into 21 different sections. However, in figure 4 only 7+7+4 (18 sections) are shown. In light of my comment above I would recommend using only 4 sections for each flow cell (12 in total).

- a general look at the data shows that for all compounds the concentration gradient goes from top to bottom. This automatically means that in the top part the compounds get diluted ($c < c_0$). This provides a potential limitation of the proposed concept, since there is a tradeoff between enrichment/dilution in the top part. The increased enrichment in the top part comes at the cost that the sample gets more

diluted with every flow cell/crack. So realistically how much can you really enrich until the concentrations become too low to do anything useful. I think this should be discussed. For example, extended figure 6 shows that serin can be enriched up to 1775x in a 20x20 flow cell network, but what would be the resulting concentration of serin if this enrichment is reached?

Referee #3 (Remarks to the Author):

Dear Editor and Authors,

I thank you for allowing me to review this new version of the article "Heat flows purify and boost reactivity of >50 building blocks of life" presented by Dr. Mast et al. In this article the authors explore, by employing microfluidic devices, the processes that occur in rock cracks, promoted by heat flows. The fundamental idea of the research is to test the capacity for selection, separation and, consequently, the concentration of organic molecules (many of them) relevant to the origin of life through said flows. The work is interesting and complete, since the experiments are complemented with numerical models. The original article already presented a very interesting perspective within the field of prebiotic chemistry, and both the method and the results constituted a novel approach in the area, capable of arousing interest in the community and encouraging academic discussion. However, some points needed to be clarified in the first version and I am pleased to note that this new version is more complete and clearer, and in this new document the small errors detected have been corrected.

The experimental design of the research was already adequate in the first version. In this new version the values and uncertainties of the measurements are presented with greater clarity. In fact, the presentation of data has been substantially improved, which is appreciated for interpreting the results. Additionally, the authors performed some of the suggested experiments that served to increase the range of their model. The most outstanding point of this research is the simulation of highly complex microenvironments to promote prebiotic chemical reactions. I really enjoyed reading the paper.

Summary of the key results

The authors make a good synthesis of the research carried out about the study subject. On the one hand, they put into context the great problem of prebiotic chemistry: the synthesis, selection and increase in concentration of some molecules. In addition, it is highlighted the need for certain conditions to be met for subsequent reactions to occur. The experimental and numerical results are presented briefly and the main findings are described too.

Originality and significance: if not novel, please include reference

As I mentioned in my previous review, the use of this type of device in prebiotic chemistry studies is very recent, and so is the use of thermal gradients in studies of separation and concentration of organic molecules. The experiments carried out are relevant, original and represent a novel contribution to the

research area since they test the separation of complex mixtures under different conditions and verify the effect of these systems on the process.

Data & methodology: validity of approach, quality of data, quality of presentation

The methodology is well established, well founded and allows experiments to be replicated, which is highly relevant, especially for those interested in the area of expertise, in which this manuscript is inserted. Further, the data are robust and are the result of several repetitions. In this type of devices, with small volumes, is expected a greater uncertainty compared to other experiments with larger volumes, this is mainly due the analytical techniques. However, the results presented in this manuscript are the result of an optimal statistical analysis.

Appropriate use of statistics and treatment of uncertainties

According to the authors, the experiments were performed in triplicate. The calculation of uncertainties and their validity are all shown in this new version in a very clear and detailed way. All values and their “absolute enrichment error” are presented in the supplementary materials.

Conclusions: robustness, validity, reliability

The conclusions are based on visibly solid results and represent a significant contribution to the field. The authors present all the results (measurements and uncertainties) of their experiments in the supplementary material. Abundantly so.

Suggested improvements: experiments, data for possible revision

All the comments and suggestions made to the first version of the manuscript have been addressed and I have no further comments for the authors whatsoever.

References: appropriate credit to previous work?

The text is correctly referenced, the respective credit is given to the work of other authors, and the most relevant articles have been selected. In fact, additional references have been included.

Clarity and context: lucidity of abstract/summary, appropriateness of abstract, introduction and conclusions

The document is clear, well written, and easy to read. The improvements, related to the representation of the data, also facilitate the reading of the figure captions.

Author Rebuttals to First Revision:

Referee #2 (Remarks to the Author):

The revised version of the manuscript includes now a connected flow chamber experiment, dimerization of glycine in a flow cell and separation of non-proteogenic amino acids as well as separation at different ionic strength. In addition, the authors provided the source data for a better understanding of their results.

In the first round of review, I suggested a validation experiment of connected flow cells, which is now shown in figure 4. However, the data does not seem robust enough to substantiate the main conclusions. I assume that there might be a problem with their quantification methodology (as outlined below).

We thank the referee for the detailed examination of our data and agree that the significance of our conclusions could be affected by the existence of the small number of outliers in our dataset. To provide strong support for our hypotheses, we therefore addressed the referee's considerations and suggestions in detail.

We repeated the measurements of the amino acids in the static heat flow chambers as well as the connected flow chamber experiments and significantly increased the data quality through a variety of improvements in the experiment as well as the HPLC analysis, as explained in detail below. With the exception of the amino acid tyrosine (Y), the thermophoretic properties (Soret coefficients) obtained matched the previous measurements to within a few percent, which proves the reproducibility of our method. Based on this data, we have updated all calculations of the grid systems - the resulting findings confirm our previous conclusions even more clearly.

1) Technically it's great that the authors run 3 repeat experiments (supplementary table 14). However, I don't understand why the authors divide the flow cells into different number of sections in each repeat (ranging from 1 to 7). Therefore, a section in one repeat is not comparable with the section of the other repeats. Why they don't keep it consistent for full comparability?

While in the previous revision, we sought to achieve maximum spatial resolution by using a high number of volume fractions, we agree with the referee that a uniform number of fractions per chamber is more reasonable here. Therefore, we repeated the experiments leading to Figure 4a-c (being detailed in the new Supplementary Figures 4+5 and Extended Data Figure 6), leaving the number of partial volumes in each repeat constant (four fractions per chamber, 12 fractions per repeat/experiment). We adjusted the values in the text accordingly, and write, for example:

Line 257-259: "The range of enrichments in a three-chamber network ($[N]/[I]$ =23-fold and $[I]/[N]$ =4.2-fold) is higher than achievable in a single chamber (Fig. 4b, left, also see Supplementary Fig. 5 for localized concentrations of individual runs)."

We hypothesized that the results in the earlier revision were more scattered due to the condensation of air moisture during freeze extraction leading to a dilution of individual samples. Also, the evaporation of very small sample volumes would lead to increasing sample concentrations. Further variations in concentration could presumably be caused by the outgassing of the solution at the temperatures used in the heat flow chambers and the resulting accumulation effects (also see <http://doi.org/10.1038/s41557-019-0299-5> on how such interfacial effects might have played a role in prebiotically plausible non-equilibrium environments).

To reduce the aforementioned artifacts as much as experimentally possible, we now perform the freeze extraction in the updated experiments embedded in a dry-ice environment to

minimize relative humidity and, thus, condensation at the extraction site. By reducing the number of extraction volumes per connected-flow-cell experiment to 4 per chamber / 12 per experiment, as suggested by the referee, we increased the sample volume to minimize evaporation and condensation effects during sampling as far as possible. In order to minimize outgassing within the heat flow chambers, we degassed the water before diluting the amino acids. We now also describe these changes in the methods section (changes marked in green):

Line 533-535: "Prior to mixing with concentrated amino acid standard, the water used for dilution was degassed by heated stirring under vacuum."

Line 541-546: "For the recovery of fractions of the three individual chambers, a slightly modified extraction procedure was used. Since the time required to extract 12 fractions (4 fractions per chamber) is increased compared to the 4 fractions used above, we added dry ice around the frozen block to reduce dilution by condensation. In addition, we directly pipetted the recovered fraction into 7 eq. of borate buffer for pre-column derivatization."

In addition to displaying the data in Supplementary Tables 14 to 16, we now also present the data values graphically in Supplementary Figure 5. The breakdown by individual repeats emphasizes the reproducibility of our results, limited only by the non-linear dependence of the accumulation on the temperature gradient (tolerance of temperature control +/-1K) and the flow rate. As detailed below, we minimize the possibility of errors in concentration analysis by using fresh HPLC columns, by extending the concentration range of the calibration measurements and by examining the best choice of HPLC detection technique (UV vs fluorescence).

2) There are large concentration differences between the repeats (sometimes up to 5-10x), but as outlined in 1) a perfect comparison cannot be made. According to the additional note, the authors attribute this to "small perturbances of the flow speed" or "generation of gas bubbles". If the authors are aware of those effects, they could have divided the flow cells into 4 instead of up to 7 sections (as they have done in Figure 2) to average out such effects.

We would like to thank the referee for these suggestions, which we have implemented as described above. In addition to the equalization of the number of extraction volumes (4 per chamber and 12 per experiment), we were able to significantly improve the data quality by also reducing the humidity during freeze extraction and degassing the solvents before use as mentioned above.

In addition, we now use thinner tubings connected to the in- and outlets to optimize the flow profiles and avoid gas leaking into the system. In the Methods section, we write:

Line 536-538: "To start the experiment, in- and outlets were connected to tubings (250 µm inner diameter to reduce air inclusions, KAP 100.966, Techlab, Germany) filled with sample solution [...]."

In addition, the authors could run the experiment at lower temperature to suppress the formation of gas bubbles. I couldn't find the actual temperature in the methods section that was used for the heating elements (top/bottom).

So far, we have only given the absolute temperature for the static heat flux chambers (line 110-111), but only the difference of temperatures for the experiments shown in Figure 4. We have corrected this and now write:

Line 539-540: "As soon as the temperature gradient (16K between 30°C and 46°C and 10K between 32°C and 42°C) was established [...]."

To further suppress the formation of gas bubbles, we choose to degas the solvent (= water) before performing the experiment as described above. This enabled us to choose the moderate temperatures of 30°C and 46°C and thus offer a most general scenario.

Most importantly, I still believe that there is a potential problem with quantification (see also 3 to 5). The authors measured calibration curves in the range of 5-75 μ M. All extreme values (supplementary table 14) seem to be outside this linear range and therefore might not be reliably quantified. As an example the authors measured two calibration curves for glycine and the equations are completely different (since they cover different ranges).

The referee is indeed right that the concentrations in the connected-flow-cell experiments sometimes significantly exceed the concentration ranges used in the calibrations. Alongside the newly performed experiments, we have now carried out the calibration measurements spanning the whole concentration range obtained in the new measurements for Figure 4a-c. We show the linear dependence between HPLC counts (integrated peaks) and concentration as well as the calibration range for each set of measurements of amino acids in Supplementary Table 4, where we list all different calibrations used in Figures 2+4, Extended Data Figures 3+4+6, Supplementary Figures, and Supplementary Tables.

In addition, for the exemplary calibrations shown at the end of the Supplementary Information, we have now included section titles (see pages 134 and 154), referencing the main text figures and detection techniques used. We added:

Page 134 (SI): "Calibration data amino acids (Fig. 2)

Fluorescence channel for all amino acids except for tryptophan and tyrosine, where the UV-channel is used (see Methods for details)."

Page 154 (SI): "Calibration data dimerization of glycine (Fig. 5)

Detection with mass spectrometry (see Methods for details)."

3) The authors ran a control experiment with $dT = 0$ (Supplementary Table 14), which shows that a gradient is not formed. However, in this experiment a single amino acid (P) is suddenly selectively depleted in repeat 1. Repeat 2 shows no significant anomalies. A look at repeat 3 shows average concentrations that are about 10x lower.

In addition, the $dT = 12$ experiment (Supplementary table 14) shows selective depletion of C. As the authors explained in their additional note even gas bubbles can't explain a selective depletion. Together, this points towards a potential problem with their quantification methodology (see also 4 and 5).

We agree with the referee that the outliers mentioned here indeed indicate a problem with the HPLC measurement. In the course of our experiments, we found that such outliers occurred particularly with older, frequently used HPLC columns, specifically in HPLC runs of amino acid mixtures. To rule out such problems in the measurements carried out for this resubmission, we used a fresh column and ran the control samples ($dT=0$) through the HPLC column at the end of the experimental sequence.

The absence of outliers in the controls, which could only be caused by problems in the HPLC runs and not by selective accumulation in the heat flow chamber, assured that the column is in good condition for all measurements up to this point.

4) The authors used the same amino acid concentrations (30uM) for the experiments shown in extended data figure 3 and 4. Repeat 1 in extended data figure 3 (Supplementary table 28) shows average concentrations between 13 and 34 uM (difference of up to 2.6x). A direct comparison of the average concentrations between extended figure 3 and 4 (Supplementary table 28 vs. 34) reveals that the values are about 3x higher for the latter (80 uM). Given that a closed flow cell and a starting concentration of 30uM was used, those values can't be correct. Of course, if there is a systematic error this might not affect the relative values but to be sure the authors should carefully check their data and quantification.

The referee rightly points out the large variations in the average concentrations in the static heat flow chambers measured for Extended Data Figures 3 and 4. As we had cleaned the HPLC fluorescence flow cell before the HPLC runs for Extended Data Figure 4, the measurement values were mistakenly increased because the old calibration was used. We now use the correct calibration (recorded after the cleaning), as listed in Supplementary Table 4. In this calibration, however, we had an overlap between amino acids G and R, which are now omitted in Extended Data Figure 4. The comparison with main text Figure 2 is still readily possible due to the remaining 15 amino acids.

As the referee has already noted, the above-mentioned residual variation of the absolute concentrations would not influence the relative enrichments in case of linear calibrations. We thank the referee for this comment and changed our analysis to linear calibrations for these experiments which proved to fit equally well (see R^2 values in Supplementary Table 4). To emphasize this, we write in the main text / methods:

Line 639-642: "For all other species, we used an external linear calibration (see Supplementary Table 3-4 and examples on pages 134-153 of the Supplementary Information). The linear calibration intrinsically emphasizes that differences in calibration or absolute concentration do not change the resulting enrichments."

To further check for systematic problems, we prepared a sample of 11 amino acids and separated it into 3 parts. These were then pre-column derivatized and analyzed as described in the Methods, observing only minor deviations between the three repeats (Supplementary Figure 15 and Supplementary Table 66).

5) Further anomalies are for example found in Supplementary Table 4 (repeat 2) where the integral of amino acid Y suddenly increases by about 20x. In repeat 3 the same amino acid gives the following integrals for section top (26,892) and II (20,093). However, the concentration given in the table is 11.807 and 25.894 uM respectively. This can't be correct. In addition, extended table 28 repeat 1 amino acid Y (section top) has an integral of 8,593 which corresponds to a concentration of 17.996. There is only one calibration curve of Y and the sample was divided into 4 sections (as in supplementary table 4). Given the linear relation of the calibration curve, something doesn't add up here.

We thank the referee for pointing this out. In the course of our measurements, we found that the fluorescence channel for tyrosine (Y) consistently yielded unreliable results and suspected the cause to be the special nature of Y in the derivatization method used (see reference 53). However, we observed that the HPLC detection of tyrosine using the UV channel (described in the Methods) yielded the expected linear dependence without the outliers noted by the referee. Accordingly, we reanalyzed the measurements shown in Figure 2 (main text) by using the UV channel of the HPLC for the detection of tyrosine (and tryptophan as before) and the fluorescence channel for all other amino acids. In one of the measurements (repeat 2), the concentration of Y could not be determined presumably due to issues in the derivatization reaction (for instance incomplete dissolution of dimers as indicated in reference 53), leading to integral values about 10x the maximal calibrated values. We therefore repeated the

experiment once more, included in Supplementary Table 6-8. The calibration coefficients for Figure 2 are given in Supplementary Table 4 and shown at the end of the Supplementary Information (page 134-153).

Since the data displayed in Figure 2 (also in Extended Data Figure 3 and Supplementary Table 6 to 8) were also used to determine the thermophoretic properties, we have updated the Soret coefficients given in Extended Data Table 1 accordingly. With the exception of tyrosine for the reasons mentioned above, the updated Soret coefficients are within the error tolerances of the old values and even deviate by less than 5% for most amino acids, which demonstrates the high reproducibility of our data.

The models used in Figure 4 (and also Extended Data Figure 9 and Supplementary Figures 6, 8, 9, 11 and 12) incorporate all/most of the amino acids and were therefore recalculated according to the updated Soret coefficients. The reaction model shown in Figure 5 does not require recalculation, as the Soret coefficient of glycine remains unchanged.

Due to the only minor changes in the Soret coefficients, all statements made in the previous manuscript still apply. We have updated the corresponding numbers in the text and marked them in green. For example, we write:

Line 300-304: "Accordingly, the maximum ratio of concentrations of the amino acid I versus N is 19-fold at higher absolute concentrations, while the ratio of N versus I in another chamber is 2000-fold at reduced absolute concentrations as shown in Fig. 4e which displays a similar but amplified pattern compared to the experiments (Fig. 4b)."

Overall, I cannot check every single value but points 2-5 are in my opinion enough to question the reliability of the data. Since the main conclusions are based on quantifications it is important that the authors carefully check their data.

We believe that the concerns rightly raised by the reviewer have been addressed by our extensive improvements to the experiments and the newly performed measurements, and that the main message of the work, the possible and available selective enrichment of prebiotic compounds from complex mixtures, is now conveyed in a much more convincing way.

Other comments:

-Why the authors jump from a closed single flow cell directly to a complex flow-connected system of three flow cells? Ideally, they would first check how the flow influences the concentration gradient in a single cell and then move to a system of interconnected flow cells. Given the presence of secondary effects ("gas bubbles" or "perturbances of the flow speed"), this would be even more important to better understand the contribution of those factors in case there is indeed no way to suppress them (see point 2).

We agree with the referee that the transition from a static heat flow chamber to a flow-connected system seems large. However, since the first chamber of this system does not depend on the subsequent chambers, it behaves like a single flow cell and, accordingly, already includes the scenario proposed by the referee.

In the context of this resubmission, it was therefore our goal to improve the quality of our measurements to such an extent that the above-mentioned problems of the entire flow-connected system (condensation, evaporation, local bubble formation) are fundamentally solved and thus the statement of this work is strengthened accordingly. We believe that we have achieved this with the above-mentioned and now implemented changes. To present the

data for the individual chambers in a most transparent way, we have now included Supplementary Figure 5 as introduced above.

- in the methods section the authors state that they divide the flow cells shown in figure 4a into 21 different sections. However, in figure 4 only 7+7+4 (18 sections) are shown. In light of my comment above I would recommend using only 4 sections for each flow cell (12 in total).

As suggested by the Referee, we now divide each partial chamber into 4 sections (see all comments above).

- a general look at the data shows that for all compounds the concentration gradient goes from top to bottom. This automatically means that in the top part the compounds get diluted ($c < c_0$). This provides a potential limitation of the proposed concept, since there is a tradeoff between enrichment/dilution in the top part. The increased enrichment in the top part comes at the cost that the sample gets more diluted with every flow cell/crack. So realistically how much can you really enrich until the concentrations become too low to do anything useful. I think this should be discussed. For example, extended figure 6 shows that serin can be enriched up to 1775x in a 20x20 flow cell network, but what would be the resulting concentration of serin if this enrichment is reached?

The referee points out an important feature of the enrichment mechanism shown in this paper. While thermophoretically stronger accumulating substances are enriched compared to those with weak thermophoresis at increased absolute concentrations (lower part of the network), the latter are enriched in the upper part of the network by a stronger depletion of the substances with strong thermophoresis.

We already pointed this out in the main text Figure 4e, for example, where the 2000-fold enrichment of asparagine (N) compared to isoleucine (I) is possible at 0.1-fold the initial concentration of asparagine. For clarity, we now show this type of scatterplot for other amino acid combinations in the new Extended Data Figure 7. In this figure, we provide the enrichments (-fold) and the corresponding serine concentrations using the specific example of isoleucine vs serine. In the main text, we additionally write:

Line 300-308: "Accordingly, the maximum ratio of concentrations of the amino acid I versus N is 19-fold at higher absolute concentrations, while the ratio of N versus I in another chamber is 2000-fold at reduced absolute concentrations as shown in Fig. 4e which displays a similar but amplified pattern compared to the experiments (Fig. 4b). Each point in Fig. 4e represents the average concentrations in the lower part of the respective heat-flow chamber of the network. Extended Data Fig. 7 shows further examples of amino acid combinations analogous to Figure 4e and demonstrates that even in the case of enrichment of the thermophoretically weaker species over the stronger one, the former occurs at significant absolute concentrations and is not diluted to trace levels."

In addition, absolute concentrations can be raised to higher levels after the heat-flow based processes discussed in this work using already known, non-selective up-concentration mechanisms (drying, enrichment at interfaces). To further clarify this point, we write:

Line 443-446: "Here, residual chemicals with weak thermophoresis are enriched at moderate thermophoretic accumulations (Extended Data Fig. 6, 7, 9). The resulting concentration ratios could be conveyed to higher absolute concentrations by non-selective up-concentration modes already discussed in the literature, such as local drying or accumulation at gas-water interfaces."

Reviewer Reports on the Second Revision:

Referees' comments:

Referee #2 (Remarks to the Author):

The revised version by Mast and colleagues, shows significant improvements in the data quality. Overall, the authors responded to all comments and tried their best to minimize any potential source of error in their experimental set up. In some repeats there are still differences of 2x. However, given the complexity of the experimental setup and analysis, a large error has to be anticipated.

Importantly, the work provides a new concept on how molecules could have been enriched and purified from relatively complex mixtures. Therefore, I support publication in Nature and congratulate the authors on their work.

Some minor comments:

- I think the title could be adjusted, as it suggests that also the reactivity of more than 50 compounds has been assessed.
- It would be great to also include the calibration curve data with a larger range (2-500 μM).
- Supplementary figure 5: In the caption it says dT 16°C, while in the figure it says dT 17°C.

Author Rebuttals to Second Revision:

Point by point response to comments of referee #2:

Referee #2 (Remarks to the Author):

The revised version by Mast and colleagues, shows significant improvements in the data quality. Overall, the authors responded to all comments and tried their best to minimize any potential source of error in their experimental set up. In some repeats there are still differences of 2x. However, given the complexity of the experimental setup and analysis, a large error has to be anticipated.

Importantly, the work provides a new concept on how molecules could have been enriched and purified from relatively complex mixtures. Therefore, I support publication in Nature and congratulate the authors on their work.

We thank the referee for the helpful comments and suggested improvements in the experiment and manuscript, which enabled us to improve the quality of our work.

Some minor comments:

- I think the title could be adjusted, as it suggests that also the reactivity of more than 50 compounds has been assessed.

We thank the referee for this assessment and agree that the current title could be misleading. We have changed it to "**Heat flows enrich prebiotic building blocks and enhance their reactivity**" in line with the editor's suggestions.

- It would be great to also include the calibration curve data with a larger range (2-500 μM).

As requested by the referee, we now also added the calibration curves with the larger range (2-500 μM), pages 155 to 165.

- Supplementary figure 5: In the caption it says dT 16°C, while in the figure it says dT 17°C.

We thank the referee for the hint and have corrected this inconsistency accordingly.